# Palmitoylated importin α regulates mitotic spindle orientation through interaction with NuMA

Patrick James Sutton ✉, Natalie Mosqueda & Christopher W Brownlee ✉

## Abstract

Regulation of cell division orientation is a fundamental process critical to differentiation and tissue homeostasis. Microtubules emanating from the mitotic spindle pole bind a conserved complex of proteins at the cell cortex which orients the spindle and ultimately the cell division plane. Control of spindle orientation is of particular importance in developing tissues, such as the developing brain. Misorientation of the mitotic spindle and thus subsequent division plane misalignment can contribute to improper segregation of cell fate determinants in developing neuroblasts, leading to a rare neurological disorder known as microcephaly. We demonstrate that the nuclear transport protein importin α, when palmitoylated, plays a critical role in mitotic spindle orientation through localizing factors, such as NuMA, to the cell cortex. We also observe craniofacial developmental defects in *Xenopus laevis* when importin α palmitoylation is abrogated, including smaller head and brains, a hallmark of spindle misorientation and microcephaly. These findings characterize not only a role for importin α in spindle orientation, but also a broader role for importin α palmitoylation which has significance for many cellular processes.

**Keywords** Spindle Orientation; Importin α; NuMA; Palmitoylation; Microcephaly
**Subject Categories** Cell Adhesion, Polarity & Cytoskeleton; Cell Cycle; Post-translational Modifications & Proteolysis

## Introduction

Mitotic spindle orientation is a fundamental cellular process which regulates cell division orientation, a crucial aspect of mitosis that when dysregulated can lead to uncontrolled asymmetric division and overproliferation, or developmental defects due to premature stem cell differentiation, depending on the cell type (Bergstralh and St Johnston, 2014; Bergstralh et al, 2017; Charnley et al, 2013; Finegan and Bergstralh, 2019). Neuronal development in particular is heavily reliant on properly oriented cell divisions to ensure the correct timing of neuron differentiation, a process that when disturbed can result in severe defects, such as microcephaly (Razuvaeva et al, 2023; Higgins and Goldstein, 2010; Konno et al, 2008).

Mitotic spindle orientation is largely controlled by the anchoring of astral microtubules (aMTs) to the cell cortex in metaphase, and the pulling force generated by the dynein/dynactin motor complex on aMTs during anaphase (Kiyomitsu and Cheeseman, 2012; Kotak and Gönczy, 2013; Toyoshima and Nishida, 2007; Pietro et al, 2016; Bergstralh et al, 2017; Singh et al, 2021; Yang et al, 2014; Anjur-Dietrich et al, 2024). aMTs are anchored at the cortex by a conserved complex of proteins consisting of Gαi, which through myristylation is thought to associate with the plasma membrane (PM), Leucine-Glycine-Asparagine repeat containing protein (LGN), which binds to Gαi, and Nuclear Mitotic Apparatus (NuMA), which binds to LGN and dynein/dynactin, facilitating the strong pulling force on the aMTs necessary for maintenance of spindle orientation in metaphase (Higgins and Goldstein, 2010; Pietro et al, 2016; Camuglia et al, 2022; Kiyomitsu and Cheeseman, 2012; Fankhaenel et al, 2023; He et al, 2023; Kiyomitsu and Boerner, 2021; Zheng et al, 2013; Okumura et al, 2018; Pirovano et al, 2019; Du and Macara, 2004; Neville et al, 2022; Yu et al, 2000; Bowman et al, 2006; Carvalho et al, 2015; Anjur-Dietrich et al, 2024). While all of these factors play a role in aMT anchoring during metaphase, only NuMA and dynein/dynactin are required during anaphase to ensure the proper separation of chromatin to the poles (Bergstralh and St Johnston, 2014; Kotak et al, 2014; Seldin et al, 2013). In addition, it has been shown that LGN and Gαi are only required for NuMA localization to the cell cortex during metaphase, not anaphase (Okumura et al, 2018; Kiyomitsu and Boerner, 2021). However, it remains unclear how these factors localize to the PM during metaphase and how this membrane localization is maintained there when it is believed that only a singly myristoylated Gαi is responsible for membrane association of the entire aMT anchoring complex. Recently it has been demonstrated that a single dynein motor interacts with each aMT, suggesting that only a single complex of Gαi, LGN, NuMA, and dynein is present at the cortex for each aMT. This highlights the need for this complex to be strongly associated with the PM (Anjur-Dietrich et al, 2024). Furthermore, the necessity for a significant pulling force towards the PM during metaphase to orient the mitotic spindle and during anaphase to facilitate separation of chromatin to the poles suggests that an additional factor, which can associate with the PM via multiple interaction points, may be involved in aMT anchoring.

Importin α is a highly conserved and abundant protein known for functioning as a nuclear transport adapter in interphase and as a spindle assembly factor during mitosis through its ability to bind nuclear localization signal (NLS) sequence containing proteins

Department of Pharmacological Sciences, Stony Brook University, Stony Brook 11794, USA. ✉E-mail: Patrick.Sutton@stonybrook.edu; Christopher.Brownlee@stonybrook.edu

(Takeda et al, 2011; Oka and Yoneda, 2018; Goldfarb et al, 2004). However in recent years, a number of proteomic screens for palmitoylated proteins utilizing a variety of biochemical methods and mass spectrometry verification have identified human importin α-1 (KPNA2) as a target for palmitoylation (Won and Martin, 2018; Thinon et al, 2018; Serwa et al, 2015; Mariscal et al, 2020; Zhou et al, 2019; Sobocinska et al, 2018; Martin and Cravatt, 2009). Palmitoylation is a reversible and dynamic process which modifies proteins post-translationally with palmitate lipids allowing both diffusion and vesicle mediated transport of palmitoylated proteins to the PM (Guan and Fierke, 2011). In addition, recent work in *Xenopus laevis* (*X. laevis*) revealed that importin α can be reversibly sequestered to the PM via palmitoylation of 4 key residues, 3 of which are conserved in humans (Brownlee and Heald, 2019). This same study found that when palmitoylated, importin α acts as an evolutionarily conserved cell-surface area-to-volume sensor that coordinately scales nuclear and spindle size to cell size. Following this work, mass spectrometry methods have confirmed palmitoylation of at least one residue in human KPNA2 in addition to the 3 residues conserved from *X. laevis* importin α-1 (Zhou et al, 2019) and palmitoylation prediction screens have identified additional cysteine residues in human KPNA2 likely to be palmitoylated with high confidence. These findings raise the intriguing possibility that importin α may have roles other than as a nuclear import adapter upon palmitoylation (Brownlee and Heald, 2019).

Due to the requirement of an exceedingly strong pulling force to drive orientation of the spindle and chromatin separation to the poles, we hypothesize that palmitoylated importin α could provide a sufficiently strong interaction with the PM to anchor aMTs in mitosis. Importin α when modified with palmitate lipids at multiple residues would provide a significantly increased membrane association compared to singly myristoylated Gαi and could increase the membrane association of the aMT anchoring complex as a whole if bound to the factors involved. Notably, NuMA as well as Discs Large (Dlg), another protein recently implicated in spindle orientation (Bergstralh et al, 2016; Carvalho et al, 2015; Saadaoui et al, 2014; Schiller and Bergstralh, 2021), contain strongly predicted NLS sequences suggesting importin α may be responsible for their cellular localization through importin α-NLS binding. Previous literature has also investigated the effects of deleting the NLS of NuMA and determined that NuMA's NLS is required for its cortical localization (Okumura et al, 2018). In addition, findings which demonstrate that NuMA does not require LGN or Gαi to localize to the polar cortex during late anaphase raise the intriguing possibility that palmitoylated importin α may play a role in NuMA's mitotic localization (Okumura et al, 2018; Kiyomitsu and Boerner, 2021). We hypothesize that palmitoylated importin α could bind NuMA's NLS in metaphase, transport it to the PM and therefore be necessary for NuMA's localization in metaphase and early anaphase, and sufficient for NuMA's localization during late anaphase.

A key factor driving protein localization and spindle formation in mitosis is a gradient of RanGTP generated by the chromosome-tethered RanGEF, RCC1 (Kalab and Heald, 2008). In interphase, the same localized concentration of RanGTP in the nucleus facilitates release of cargo from importins for proper nuclear transport (Kalab and Heald, 2008; Ozugergin and Piekny, 2021). All of these processes are driven by the binding of RanGTP to importin β1, which causes the dissociation of importins and any

bound cargo (Kalab and Heald, 2008; Oka and Yoneda, 2018). In mitosis, chromosomes are arranged during metaphase in such a manner that the tethered RCC1 generates a high concentration of RanGTP at the midline of the cell, but a low concentration of RanGTP towards the poles (Ems-McClung et al, 2020). The lack of RanGTP at the polar cortex and the abundance of RanGTP at the lateral cortex during metaphase due to the equatorial localization of the mitotic Ran gradient indicates that importin α can remain bound to NLS containing cargo at the polar cortex exclusively. Importantly, this is where aMTs are anchored and NuMA has been found to localize (Kiyomitsu and Cheeseman, 2012; Oka and Yoneda, 2018; Chang et al, 2017; Kalab and Heald, 2008; Ems-McClung et al, 2020). This suggests that the Ran gradient can regulate the localization of NLS containing proteins at the PM and therefore restrict localization of NuMA to the polar cortex, through interaction with palmitoylated importin α. In addition, it has been shown that manipulation of the Ran gradient disrupts spindle orientation and the localization of spindle orientation factors (Kiyomitsu and Cheeseman, 2013).

In the present work, we demonstrate that importin α when palmitoylated is a key regulator of mitotic spindle orientation through localization of NuMA to the PM during metaphase and as such palmitoylation of importin α is required for proper control of cell division orientation. We show that palmitoylated importin α can localize throughout the PM in mitotic cells and that palmitoylation of importin α is required for both proper spindle orientation and proper NuMA localization in metaphase. While importin α localizes to the entire PM when palmitoylated (at both the lateral and polar cortex), the accumulation of RanGTP near the metaphase plate would preclude importin α binding to NuMA, leading to accumulation of NuMA specifically to the polar cortex, where aMTs are anchored. We also explore the effects of importin α palmitoylation disruption on neuronal development in *X. laevis* and observe microcephaly which can be rescued upon forcing importin α to the PM independent of palmitoylation, further supporting that PM localization of palmitoylated importin α is a key regulator of mitotic spindle orientation.

## Results

### Importin α localizes to the mitotic polar cortex at metaphase in a palmitoylation-dependent, but not cargo-dependent manner

Importin α has long been recognized for its role in spindle assembly during mitosis by facilitating the transport of spindle assembly factors to the midline of the cell, where mitotic spindles form (Kalab and Heald, 2008; Kaláb et al, 2006; Weaver and Walczak, 2015; Goldfarb et al, 2004). Furthermore, it has been demonstrated to influence spindle orientation through regulation of TPX2 activity (Guo et al, 2019, 2021). However, recent work has demonstrated in *X. laevis* that importin α can be post-translationally modified with palmitate lipids to drive PM localization (Brownlee and Heald, 2019). These palmitoylation sites are conserved in human importin α and additional palmitoylation sites have been confirmed by palmitoylation prediction screens (Fig. EV1A) and multiple mass spectrometry studies of global palmitoylated proteins (Mariscal et al, 2020; Zhou et al, 2019; Won and Martin, 2018; Thinon et al,

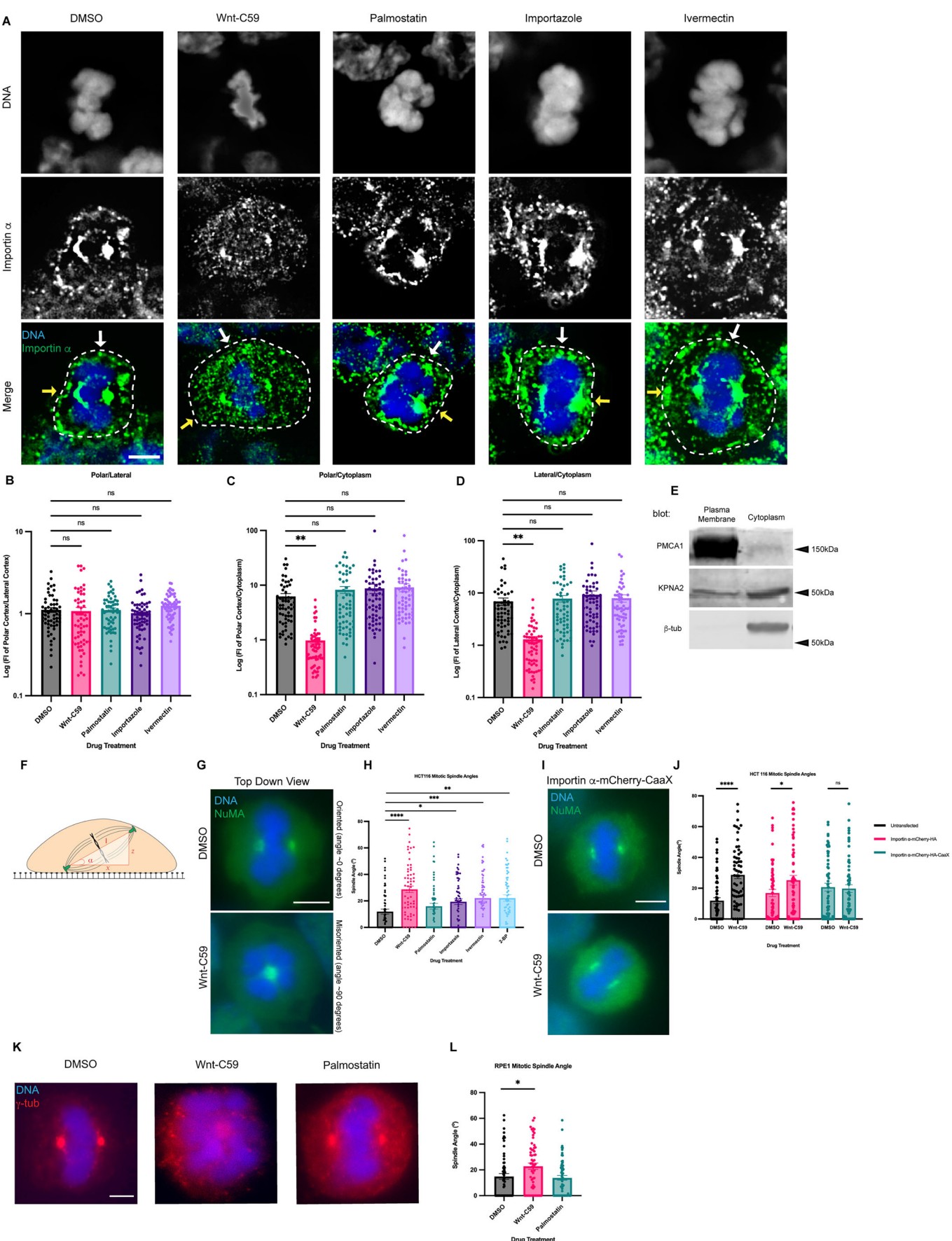

◄ **Figure 1. Palmitoylation mediated cortical localization of importin α and importin α cargo binding are required for proper mitotic spindle orientation.**

(A) Immunofluorescence images of importin α localization in metaphase-arrested (refer to Methods) HCT116 cells incubated for 1 h with DMSO, drugs that inhibit importin palmitoylation (10 μM Wnt-C59) drugs that enhance importin palmitoylation (50 μM palmostatin), and drugs that inhibit importin cargo binding (25 μM ivermectin) or cargo release (40 μM importazole). Yellow arrows indicate cortical poles and white arrows indicate lateral cortex. Scale bar = 5 μm. Cell boundaries determined by brightfield. (B–D) Quantification of importin α localization in drug treated cells. Measurements of importin α signal intensity were made at three cellular locations: polar cortex, lateral cortex and cytoplasm. Polar cortex measurements were made for each cell at the pole with the higher measure of intensity, a similar method was used for lateral cortex measurements and cytoplasm measurements were made at the midline of the cell. These measurements were normalized to each other on a cell-by-cell basis by determining the ratio of cortical vs lateral importin α, cortical vs cytosolic importin α, and lateral vs cytosolic importin α. Mean +/− SEM n = 60 from 2 biological replicates. (C) **p = 0.0093, (D) **p = 0.0029 determined by Student's t-test. (E) Western blot of HCT116 cell fractions following subcellular fractionation to separate PM, cytoplasmic, organelle, and nuclear fractions. PM and cytoplasmic fractions shown. Western blots were immunostained for PMCA1 (PM marker), β-tubulin (cytoplasmic marker), and importin α (KPNA2). (F) Cartoon representation of a metaphase cell with misoriented spindles mounted on a coverslip indicating the angle, α, which was measured as the arctangent of the horizontal distance, z, over the vertical distance, x, between the two centrosomes to determine the angle of spindle structures. (G) Immunofluorescence images of metaphase-arrested HCT116 cells in presence of DMSO or 10 μM Wnt-C59 stained for NuMA. DMSO treated cell represents a properly oriented cell with a spindle angle of 0 degrees relative to the parallel of the coverslip the cells were mounted on. Wnt-C59 treated cell represents a severely misoriented cell with a spindle angle of 90 degrees relative to the parallel of the coverslip the cells were mounted on. Scale bar = 5 μm. (H) Quantification of mitotic spindle angles for metaphase-arrested HCT116 cells incubated in DMSO, 10 μM Wnt-C59, 50 μM palmostatin, 40 μM importazole, 25 μM ivermectin, or 100 μM 2-bromopalmitate for 1 h prior to analysis. All drug treatments except palmostatin significantly increased the mean spindle angle of metaphase cells relative to a DMSO control. Mean +/− SEM, n = 60 mitotic cells from 2 biological replicates. *p = 0.0103, **p = 0.0012, ***p = 0.0009, ****p < 0.0001 determined by Student's t-test. Refer to Methods for method of determining spindle angle. (I) Immunofluorescence images of metaphase-arrested HCT116 cells transfected via nucleofection with importin α-mCherry-HA-CaaX and incubated in either DMSO or 10 μM Wnt-C59 for 1 h prior to analysis stained for NuMA. Scale bar = 5 μm. (J) Quantification of mitotic spindle angles for metaphase-arrested HCT116 cells incubated with DMSO or 10 μM Wnt-C59 expressing importin α-mCherry-HA or importin α-mCherry-HA-CaaX. Cells expressing importin α-mCherry-HA-CaaX showed no spindle misorientation when treated with Wnt-C59. Mean +/− SEM, n = 60 mitotic cells from 2 biological replicates *p = 0.0134, ****p < 0.0001 determined by Student's t-test. (K) Immunofluorescence images of metaphase-arrested RPE-1 cells incubated with DMSO, 10 μM Wnt-C59, or 50 μM palmostatin for 1 h prior to analysis stained for γ-tubulin. DMSO treated cells are representative of cells with properly oriented spindles. Wnt-C59 treated cells were significantly misoriented compared to DMSO control. Palmostatin treated cells were properly oriented when compared to DMSO control. Scale bar = 5 μm. (L) Quantification of mitotic spindle angles for metaphase-arrested RPE-1 cells incubated with DMSO, 10 μM Wnt-C59 or 50 μM palmostatin for 1 h prior to analysis. Wnt-C59 treatment significantly increased the mean spindle angle of metaphase cells relative to a DMSO control while palmostatin treatment did not significantly increase the mean spindle angle relative to a DMSO control. Mean +/− SEM, n = 60 mitotic cells from 2 biological replicates. *p = 0.0175 determined by Student's t-test. Source data are available online for this figure.

2018; Serwa et al, 2015; Sobocinska et al, 2018). In addition, NLS prediction screens have shown an enrichment of NLS containing proteins which localize to the PM and play a role in a variety of cellular processes, including spindle orientation (Fig. EV1B,C). This suggests that palmitoylated importin α may be involved in mitotic cellular processes outside of spindle assembly at the PM by binding NLS containing proteins and localizing them there.

We sought to investigate other possible roles of palmitoylated importin α in mitosis by either increasing or decreasing levels of palmitoylated importin α, as well as inhibiting importin α's ability to bind or release cargo. To manipulate palmitoylation of importin α, we targeted the specific proteins responsible for palmitoylating and depalmitoylating importin α. Palmitoylation is catalyzed by palmitoyl acyl transferases (PATs), which attach palmitoyl groups to serine or cysteine residues, while depalmitoylation is carried out by acyl protein thioesterases (APTs) which remove palmitoyl groups (Guan and Fierke, 2011). To modulate palmitoylation levels specifically, we used small molecule inhibitors to target porcupine (PORCN), the serine PAT responsible for palmitoylation of serine residues on importin α, and APT1, the APT responsible for depalmitoylation of importin α (Brownlee and Heald, 2019). HCT116 colorectal cancer cells were selected for this study due to their established role as a model system for both spindle orientation and NuMA localization (Okumura et al, 2018; Tsuchiya et al, 2021).

We investigated the role of importin α in palmitoylation-mediated PM targeting using HCT116 cells synchronized in metaphase and treated with either DMSO control or Wnt-C59, a competitive inhibitor of PORCN (Proffitt et al, 2013). Wnt-C59 treatment would be expected to result in a reduction in cellular palmitoylated importin α levels, leading to a decrease in the population of importin α localized to the PM. Conversely,

palmostatin treatment, which inhibits APT1 (Dekker et al, 2010; Lin and Conibear, 2015), would be expected to result in an increase in palmitoylated importin α levels, known as hyper-palmitoylation, which would increase the population of importin α localized to the PM.

Mitotic importin α cellular localization was assessed after 1 h in the respective drug treatments, with cell boundaries determined by bright-field microscopy (Fig. EV2A). In DMSO control cells, importin α is observed localizing to centrosomes, near the assembling spindles, and at the PM (Fig. 1A). Notably, Wnt-C59 treatment resulted in a 6-fold decrease of importin α signal at the PM and a concurrent enrichment of importin α signal in the cytoplasm (Fig. 1A–D), while palmostatin treatment did not alter importin α PM localization compared to DMSO control treatment (Fig. 1A–D). PM localization in DMSO control cells was confirmed by western blot analysis of subcellular-fractioned HCT116 cells. HCT116 cells were metaphase arrested and fractionated into 4 fractions: nuclear, cytoplasmic, organelles, and PM. The PM and cytoplasmic fractions were analyzed by SDS-PAGE and immunoblotted for β-tubulin, as a cytoplasmic marker, plasma membrane Ca²⁺ ATPase1 (PMCA1), as a PM marker, and importin α. β-tubulin was detected only in the cytoplasmic fraction and PMCA1 was detected only in the PM fraction demonstrating proper isolation of cellular fractions. However, importin α was detected in both the cytoplasmic and PM fractions confirming our immunofluorescence findings that importin α localizes to the PM in metaphase-arrested cells (Fig. 1E). These results collectively suggest that palmitoylation of importin α during metaphase is required for its proper localization to the cortex.

The localization of importin α during metaphase was also analyzed in the presence of drugs that disrupt its ability to bind and

release cargo. Metaphase-arrested HCT116 cells were treated with either ivermectin, a small molecule inhibitor that prevents the binding of importin α to NLS containing cargoes (Wagstaff et al, 2012) or importazole, a small molecule inhibitor of importin based nuclear transport that prevents RanGTP mediated release of cargo from importins (Soderholm et al, 2011) (Fig. 1A–D). We observed that in both importazole and ivermectin treated cells, importin α localization to the PM remained unperturbed (Fig. 1A–D). Notably, in all drug treatments mitotic spindles were still formed properly despite disruption of importin α palmitoylation or NLS cargo binding. Taken together, these results suggest importin α partitions to the mitotic cortex via palmitoylation and does not require binding of NLS-containing cargoes to localize to the PM.

It is noteworthy that we also observe alterations in chromatin size in metaphase cells treated with our drug array. Specifically, palmostatin, importazole and ivermectin treatments appear to induce larger, more condensed chromatin compared to that observed in DMSO or Wnt-C59 treated cells. This is in line with what has previously been reported by Brownlee and Heald in 2019 and Zhou et al in 2023. Specifically, palmitoylated importin α has been shown to regulate organelle size scaling (Brownlee and Heald, 2019) and chromatin size scaling (Zhou et al, 2023).

## Importin α palmitoylation and cargo binding is required for proper mitotic spindle orientation

We next sought to determine the effect of both disrupting importin α's ability to properly bind NLS-containing cargo and importin α mislocalization on the orientation of the metaphase mitotic spindle. HCT116 cells were metaphase-arrested and treated with DMSO, Wnt-C59, palmostatin, importazole, ivermectin, or 2-bromopalmitate, a pan palmitoylation inhibitor (Lin and Conibear, 2015). The angle of the metaphase spindle was then calculated using the relative horizontal and vertical distances between the two spindle poles (Fig. 1F). Mitotic spindles were found to be misoriented in all treatments other than palmostatin relative to the DMSO control (Fig. 1G,H). Interestingly, disruption of importin α palmitoylation (Wnt-C59 treatment), which was previously shown to alter mitotic importin α localization, and disruption of importin α's ability to properly bind and release cargo (ivermectin and importazole treatments, respectively) both resulted in spindle misorientation. These data suggest that importin α palmitoylation is required for proper spindle orientation, and spindle orientation is dependent upon importin α binding of NLS-containing cargo.

To determine if the spindle misorientation phenotypes observed when HCT116 cells were treated with palmitoylation disrupting drugs were specifically due to mislocalization of importin α from the cell cortex, we constructed a plasmid containing importin α modified with a C-terminal CaaX domain. Addition of the CaaX domain to importin α facilitates its cortical localization through farnesylation, irrespective of its palmitoylation status (Fig. EV3). Unlike palmitoylation, farnesylation is irreversible, enabling CaaX-modified proteins to be directed to the membrane and remain tethered there (Tang et al, 2009; Tamanoi et al, 2001). HCT116 cells were transfected via nucleofection to express CaaX-modified importin α or an unmodified importin α 24 h prior to metaphase arrest and subsequent drug treatment with DMSO or Wnt-C59. We observed that HCT116 cells overexpressing unmodified importin α exhibited significantly misoriented spindle structures when treated

with Wnt-C59 compared to DMSO treatment (Fig. 1J). However, HCT116 cells overexpressing CaaX-modified importin α did not exhibit misorientation of spindle structures under the same conditions (Fig. 1I,J), therefore rescuing spindle misorientation in Wnt-C59 treatment. This suggests that the spindle misorientation phenotypes observed when palmitoylation is disrupted are specifically due to mislocalization of importin α to the mitotic cell cortex.

Several cancer cell lines have been shown to harbor various mutations which exacerbate mitotic spindle orientation phenotypes (Chhabra and Booth, 2021). Therefore, we repeated the previous spindle orientation experiment using an hTERT immortalized cell line, RPE-1, treated with Wnt-C59 and palmostatin (Fig. 1K). We again observed that Wnt-C59, but not palmostatin treatment resulted in mitotic cells with significantly misoriented spindle structures relative to the DMSO control (Fig. 1L). This confirmed the findings in HCT116 cells and further suggests that palmitoylation of importin α is required for proper mitotic spindle orientation.

Previous research has indicated that cell shape and stretch play a role in determining the axis of spindle orientation and, consequently, division orientation (Finegan and Bergstralh, 2019; Charnley et al, 2013). While recent work has demonstrated that mechanical force applied to cells through stretching has a more pronounced effect on spindle orientation (by directly influencing NuMA localization) than cell shape (Tarannum et al, 2022), we sought to determine whether our palmitoylation and importin function-disrupting drug array had an impact on mitotic cell shape. We measured the circularity of 60 mitotic HCT 116 cells per drug treatment and observed that there was no significant reduction in cell circularity in any of our drug treatments (Fig. EV2B). In fact, palmostatin treated cells exhibited a slightly increased circularity compared to DMSO treated cells (Fig. EV2B). These findings confirm that the observed spindle misorientation is not attributable to alterations in cell shape.

## Importin α interacts with NuMA but not Dlg at the metaphase cell cortex

Mitotic spindle orientation in vertebrates is mediated through aMT anchoring at the cell cortex via a conserved protein complex consisting of LGN, Gαi, NuMA, and dynein/dynactin (Bergstralh et al, 2017). Of these known spindle orientation proteins, only NuMA contains a known NLS sequence, suggesting it may be a potential binding partner of importin α. To confirm the interaction between importin α and NuMA, we performed immunoprecipitation of the endogenous proteins from metaphase-arrested HCT116 cells treated with DMSO, Wnt-C59, and palmostatin. Our results demonstrated that importin α and NuMA coprecipitated equally in all drug treatment conditions (Fig. 2A), confirming importin α and NuMA binding which is unaffected by palmitoylation disruption.

To further investigate if importin α-NuMA interactions occur specifically at the mitotic cortex, where aMTs are anchored, we employed a rolling amplification-based proximity ligation assay (PLA) (Alam, 2018). In addition, we also probed for importin α-Dlg interactions at the mitotic cortex using the same assay, as Dlg has been recently implicated in spindle orientation (Schiller and Bergstralh, 2021; Saadaoui et al, 2014; Bergstralh et al, 2016), and contains a predicted NLS sequence. Therefore, we also probed for importin α-Dlg

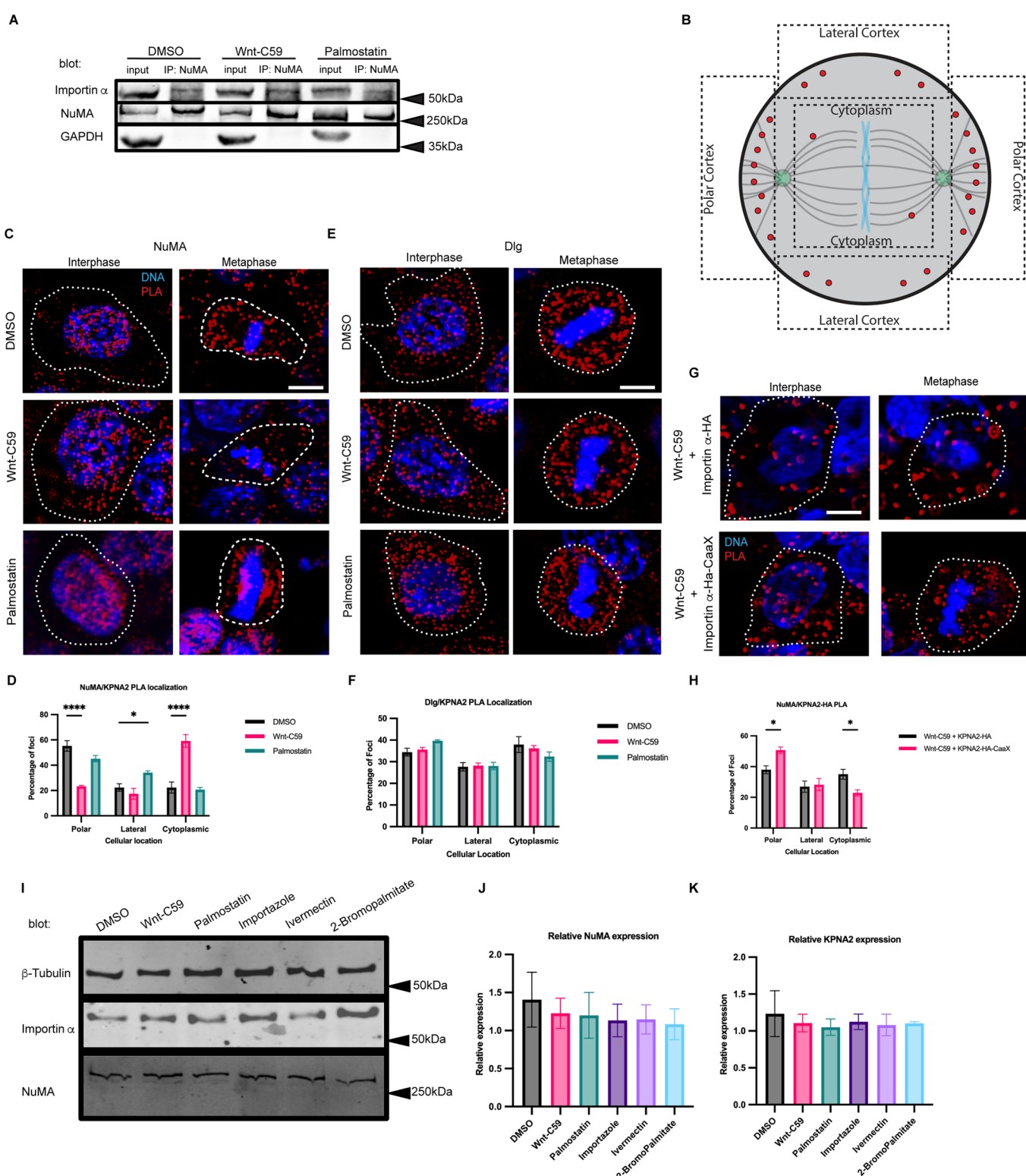

interactions at the mitotic cortex using the same assay. Importin α interaction with NuMA or Dlg was quantified by measuring the number of PLA foci in 3 regions of interest at the polar cortex, lateral cortex, and in the cytoplasm to determine where in the cell these proteins were interacting (Fig. 2B). PLA was performed on metaphase-arrested HCT116 cells in DMSO control, Wnt-C59 and palmostatin treated conditions using specific antibodies to importin α and either NuMA or Dlg (Fig. 2C and E, respectively). The assay revealed that importin α and NuMA interact at the polar cortex of DMSO treated metaphase cells, but not Wnt-C59 treated cells. In addition, importin α-NuMA interactions

**Figure 2.  Importin α interacts with NuMA, but not Dlg at the metaphase cell cortex in a palmitoylation-dependent manner.**

(A) Western blot of NuMA immunoprecipitation from HCT116 cells treated with DMSO, 10 μM Wnt-C59 or 50 μM palmostatin for 1 h prior to analysis. Immunoprecipitation of NuMA followed by importin α and NuMA western blot. (B) Schematic of PLA quantification. ROIs of quantification represented by dashed lines. (C) Immunofluorescence images of DuoLink proximity ligation assay (PLA) probing interaction of NuMA with importin α (KPNA2) in interphase and metaphase-arrested HCT116 cells in the presence of DMSO, 10 μM Wnt-C59 or 50 μM palmostatin for 1 h prior to analysis. White dashed lines indicate cell borders as determined by brightfield images. Scale bar = 5 μm. (D) Quantification of the percentage of importin α (KPNA2)-NuMA PLA foci at the polar cortex, lateral cortex and cytoplasm in DMSO, Wnt-C59 and palmostatin treated cells. Foci were enriched at the polar cortex in DMSO treated cells, the cytoplasm in Wnt-C59 treated cells, and the lateral cortex in palmostatin treated cells. Mean +/− SEM, n > 136 foci, *p = 0.0132, ****p < 0.0001 determined by Student's t-test. (E) Immunofluorescence images of DuoLink PLA probing interaction of Dlg with importin α (KPNA2) in interphase and metaphase-arrested HCT116 cells in the presence of DMSO, 10 μM Wnt-C59 or 50 μM palmostatin. White dashed lines indicate cell borders. Scale bar = 5 μm. (F) Quantification of the percentage of importin α (KPNA2)-Dlg PLA foci at the polar cortex, lateral cortex and cytoplasm in DMSO, Wnt-C59, and palmostatin treated cells. Localization of PLA foci did not change across three drug treatments. Mean +/− SEM, n > 297 foci, all data points are non-significant as determined by Student's t-test. (G) Immunofluorescence images of DuoLink proximity ligation assay probing interaction of NuMA with nucleofected importin α constructs (KPNA2-HA-mCherry and KPNA2-HA-mCherry-CaaX) in interphase and metaphase-arrested HCT116 cells in the presence of DMSO or 10 μM Wnt-C59 for 1 h prior to analysis. White dashed lines indicate cell borders as determined by brightfield images. Scale bar = 5 μm. (H) Quantification of the percentage of nucleofected importin α constructs (KPNA2)-NuMA PLA foci at the polar cortex, lateral cortex and cytoplasm in Wnt-C59 treated cells. Cells nucleofected with importin α-HA and treated with Wnt-C59 did not exhibit enrichment of foci at the PM and instead showed foci throughout the cell. Cells nucleofected with importin α-HA-CaaX and treated with Wnt-C59 exhibited foci enrichment at the PM indicating that expression of importin α-HA-CaaX rescued the effects of Wnt-C59 treatment. Mean +/− SEM, n > 195 foci. Polar *p = 0.0279, lateral *p = 0.0393 determined by Student's t-test. (I) Western blot of importin α, NuMA and β-tubulin in metaphase-arrested HCT116 cells in the presence of DMSO, 10 μM Wnt-C59, 50 μM Palmostatin, 40 μM importazole, 25 μM ivermectin, or 100 μM 2-bromopalmitate. (J, K) Quantification of NuMA and importin α (KPNA2) expression levels, respectively, relative to β-tubulin protein levels for each condition. Mean +/− SEM, n = 3. Source data are available online for this figure.

were found at both the polar and lateral cortices in palmostatin treated cells (Fig. 2D). Palmostatin treated cells also exhibited a marked enrichment of importin α-NuMA interactions at the centrosomes and along spindle structures (Fig. 2D) suggesting that importin α could associate more strongly with NuMA when hyper-palmitoylated. It is noteworthy that importin α and NuMA were only found to interact at the polar cortex in control conditions as we have previously demonstrated that palmitoylated importin α localizes throughout the mitotic PM at both the polar and lateral cortex (Fig. 1A). This is to be expected, as although importin α is present throughout the cortex, the presence of the RanGTP gradient at the midline of the cell prevents importin α from binding or remaining bound to the NLS of NuMA at the lateral cortex, limiting their interaction to the polar cortex where aMTs are anchored.

When probing for importin α interaction with Dlg, we observed that while importin α and Dlg were found to interact throughout the cytosol, this interaction was not enriched at any specific cellular location and was not altered by Wnt-C59 or palmostatin treatment (Fig. 2F). These results collectively suggest that precise importin α palmitoylation levels are required to maintain the importin α-NuMA interaction at the mitotic polar cortex and that importin α and Dlg do not interact at the PM. A potential explanation for importin α interacting with NuMA specifically and not Dlg is that these proteins contain distinct NLS sequences, such that NuMA has a bipartite NLS while Dlg is predicted to have a monopartite NLS (Chang et al, 2017). It is plausible that palmitoylated importin α, which would need to bind to these proteins in order to localize them to the membrane, binds preferentially to some NLS containing cargo over others.

To further validate that the interaction between NuMA and importin α occurs at the polar cortex in a palmitoylation-dependent manner, we transfected HCT116 cells with either importin α-mCherry-HA or importin α-mCherry-HA-CaaX. As before, we treated the cells with Wnt-C59, arrested them in metaphase, and performed a PLA for NuMA and the transfected construct (Fig. 2G). We observed that cells transfected with the wild-type importin α construct treated with Wnt-C59 (Fig. 2H) exhibited a PLA foci distribution similar to endogenous importin α expressing cells treated with Wnt-C59 (Fig. 2C). Conversely,

cells transfected with the importin α-CaaX construct treated with Wnt-C59 displayed a PLA foci distribution more similar to DMSO control cells, with a significant shift in foci from the cytoplasm to the polar cortex compared to the wild-type transfected cells. This shift suggests a rescue of importin α-NuMA interactions solely by forcing importin α to the PM (Fig. 2H).

To account for potential differences in NuMA or importin α protein levels after drug treatments, we performed a western blot and found that none of the treatments altered protein levels in metaphase-arrested HCT116 cells (Fig. 2I–K). Taken together, these results suggest that importin α interacts with NuMA at the mitotic polar cortex in a palmitoylation-dependent manner.

## NuMA and dynein/dynactin localization during metaphase is dependent on NuMA binding to palmitoylated importin α

It has previously been shown that deletion of NuMA's NLS causes mislocalization of NuMA away from the polar cortex (Okumura et al, 2018). Based on this and our findings that importin α palmitoylation is required for proper spindle orientation, we reasoned that importin α may be driving NuMA's mitotic localization to the polar cortex through binding NuMA's NLS and anchoring it to the PM. We therefore sought to determine how disruption of importin α palmitoylation and its ability to properly bind and release cargo could affect NuMA's mitotic localization.

Metaphase-arrested HCT116 cells expressing mClover-NuMA were treated with palmitoylation and importin α cargo binding altering drugs and analyzed for NuMA localization in metaphase (Fig. 3A). NuMA intensity was measured at three cellular locations (polar cortex, lateral cortex, and cytoplasm). To account for cell-to-cell differences in fluorescent intensity these measurements were normalized by calculating the ratio of polar cortex/lateral cortex, polar cortex/cytoplasm, and lateral cortex/cytoplasm fluorescent intensities. In the DMSO control there is an enrichment of NuMA at the polar cortex over the lateral cortex which is consistent with previously reported NuMA localization demonstrating that mClover-NuMA expression has no impact on localization

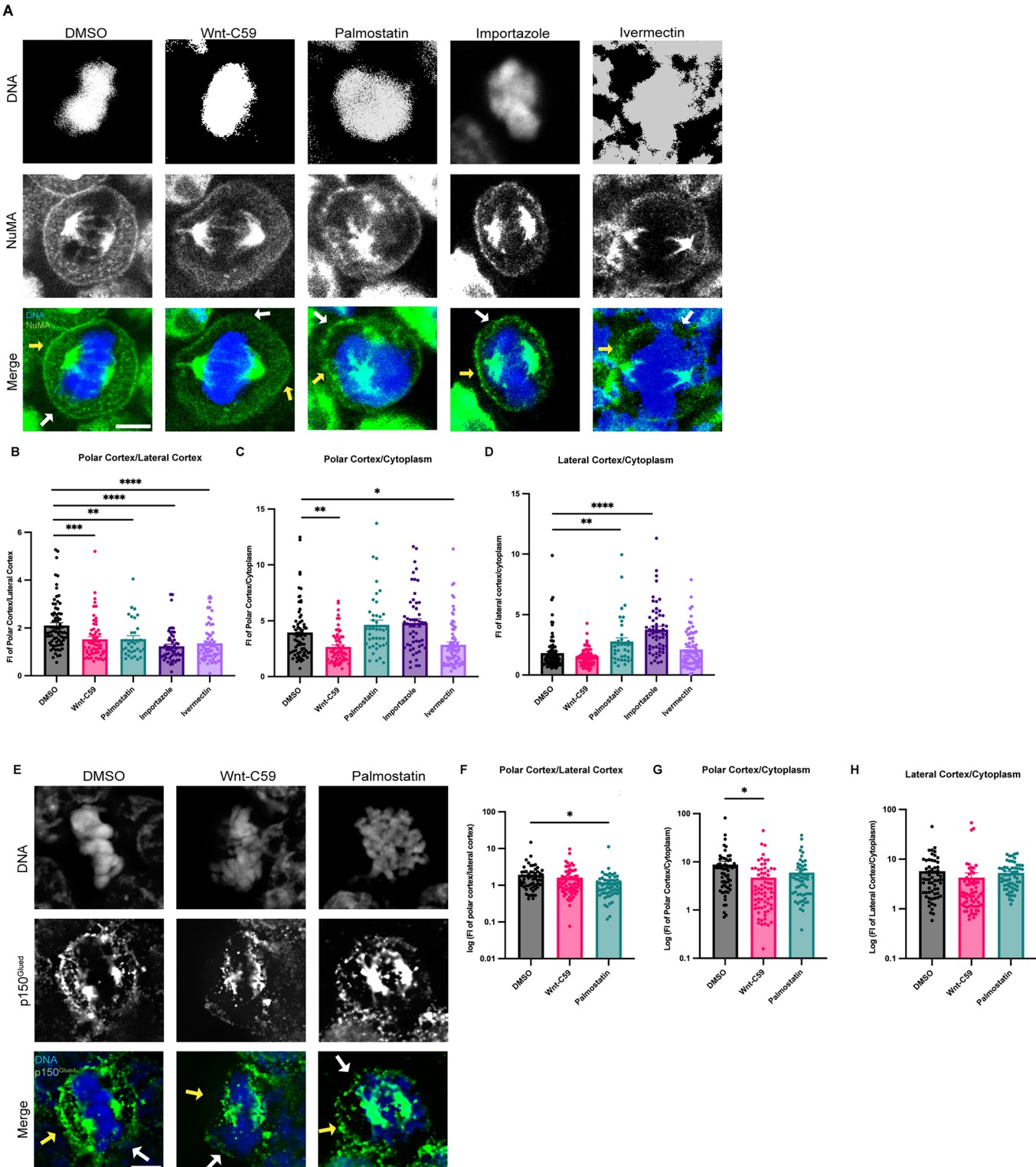

(Kiyomitsu and Boerner, 2021). It is worth noting that while importin α localizes to the PM at both the lateral and polar cortices (Fig. 1A), NuMA only localizes to the polar cortex in control conditions (Fig. 3A). This is expected as the RanGTP gradient at the midline of the cell would prevent NLS containing cargo, such as NuMA, from remaining bound to importin α at the lateral cortex. Strikingly, NuMA polar cortex enrichment was lost in all drug conditions (Fig. 3B). In Wnt-C59 and ivermectin treated cells, NuMA localization shifts away from the polar cortex and is instead enriched in the cytoplasm compared to the DMSO control (Fig. 3C).

**Figure 3.   Palmitoylated importin α regulates cortical localization of NuMA and dynein/dynactin in metaphase.**

(A) Confocal images of NuMA localization in metaphase-arrested HCT116 cells in the presence of DMSO, 10 μM Wnt-C59, 50 μM palmostatin, 40 μM importazole or 25 μM ivermectin. Yellow arrows indicate cortical poles and white arrows indicate lateral cortex. Scale bar = 5 μm. (B–D) Quantification of NuMA localization in drug treated cells. Measurements of NuMA signal intensity were made at three cellular locations: polar cortex, lateral cortex and cytoplasm. Polar cortex measurements were made for each cell at the pole with the higher measure of intensity, a similar method was used for lateral cortex measurements and cytoplasm measurements were made at the midline of the cell. These measurements were normalized on a cell-by-cell basis by determining the ratio of polar vs lateral NuMA, polar vs cytosolic NuMA, and lateral vs cytosolic NuMA. Mean +/− SEM $n > 40$. (B) **$p = 0.0064$, ***$p = 0.0005$, ****$p < 0.0001$, (C) *$p = 0.0142$, **$p = 0.0047$, (D) **$p = 0.0047$, ****$p < 0.0001$ determined by Student's t-test. (E) Immunofluorescence images of p150$^{Glued}$ localization in metaphase-arrested HCT116 cells treated with either DMSO, 10 μM Wnt-C59, or 50 μM palmostatin. Yellow arrows indicate cortical poles and white arrows indicate lateral cortex. Scale bar = 5 μm. (F–H) Quantification of p150$^{Glued}$ localization in drug-treated cells. Measurements of p150$^{Glued}$ signal intensity were made at three cellular locations: polar cortex, lateral cortex and cytoplasm. Polar cortex measurements were made for each cell at the pole with the higher measure of intensity, the same method being used for lateral cortex measurements and cytoplasm measurements at the midline of the cell. These measurements were normalized on a cell-by-cell basis by determining the ratio of polar vs lateral p150$^{Glued}$, polar vs cytosolic p150$^{Glued}$, and lateral vs cytosolic p150$^{Glued}$. Mean +/− SEM $n = 60$. (F) *$p = 0.0336$, (G) *$p = 0.0112$ determined by Student's t-test. Source data are available online for this figure.

In palmostatin and importazole treated cells, NuMA localization was no longer enriched at the polar cortex, but rather localized throughout the cortical membrane, including the lateral cortex (Fig. 3D). This unexpected lateral cortex localization when importin α is hyper-palmitoylated upon palmostatin treatment, or cannot release cargo upon importazole treatment, suggests that importin α may be an upstream regulator of proper NuMA localization at the polar cortex. The observed NuMA localization in palmostatin treated cells is of particular interest, as there was no significant spindle misorientation observed under these conditions, yet NuMA mislocalizes nonetheless. We hypothesize that the complete absence of NuMA at the cell cortex as observed in Wnt-C59 treated cells has a more pronounced effect on spindle misorientation than NuMA localizing to both polar and lateral cortices, as seen in palmostatin treated cells (Fig. 3A). In the conditions with no cortical NuMA, aMTs would not be anchored to the cell cortex at all, while in conditions with NuMA present at both the polar and lateral cortices, aMTs would still anchor to the cortex albeit without spatial regulation. These results of NuMA mislocalization upon palmitoylation and importin α function disruption suggest that palmitoylated importin α membrane localization and cargo binding is necessary for NuMA's localization and maintenance to the mitotic polar cortex.

As dynein/dynactin are known to associate with the aMT anchoring complex through NuMA binding, and we have found that palmitoylated importin α is required for NuMA PM localization, we investigated the impact of disrupting importin α palmitoylation on dynactin localization in metaphase-arrested HCT116 cells. Metaphase-arrested cells were treated with DMSO, Wnt-C59 or palmostatin as described previously. We visualized p150$^{Glued}$, the large subunit of dynactin to analyze localization of dynein/dynactin (Fig. 3E). Dynactin localization was quantified using the same method employed for NuMA localization. We found that in DMSO treated control cells dynactin was enriched at the polar cortex relative to the lateral cortex and was substantially localized to the cortex relative to the cytoplasm (Fig. 3F–H). This localization pattern closely resembles that of NuMA and was similarly disrupted by both Wnt-C59 and palmostatin treatments, albeit in distinct manners. In Wnt-C59 treated cells, dynactin exhibited a significant increase in the cytoplasm relative to the polar cortex (Fig. 3G). Conversely, in palmostatin treated cells, dynactin showed a significant increase at the lateral cortex relative to the polar cortex (Fig. 3F). This pattern of mislocalization largely aligns with that of NuMA, suggesting that palmitoylated importin α

is driving NuMA-dynein/dynactin localization to the polar cortex, facilitating aMT anchoring. Notably, unlike NuMA, dynactin did not exhibit a significant enrichment at the lateral cortex relative to the cytoplasm in palmostatin treated cells (Fig. 3H). This disparity in localization patterns could be attributed to various factors that would preclude excess dynein/dynactin from accumulating at the lateral cortex, where the number of aMTs is significantly lower compared to the polar cortex. Further investigation is warranted to elucidate the underlying mechanisms responsible for this disparity. Nevertheless, the overall pattern of misorientation when importin α palmitoylation is disrupted strongly supports the role of palmitoylated importin α role in NuMA-dynein/dynactin localization.

## Disruption of importin α palmitoylation results in microcephaly-associated phenotypes in *X. laevis*

Having established the role of palmitoylated importin α in mitotic spindle orientation by localizing NuMA to the polar cortex and maintaining it there, we subsequently sought to validate these findings in vivo. Numerous centrosomal and mitotic spindle orientation defects have been linked to microcephaly, as the resulting misorientation of cell division during brain development leads to a depletion of neuroprogenitor cells, and consequently, total brain tissue at the completion of development (Bergstralh and St Johnston, 2014). The African clawed frog *Xenopus laevis* has been used extensively as a model system to study microcephaly and developmental craniofacial abnormalities (Kennedy and Dickinson, 2014; Lasser et al, 2019; Shantanam and MUELLER, 2018). *X. laevis* has also been heavily utilized in mitotic spindle orientation studies exploring the roles of cell shape and stretch on NuMA localization and in turn spindle orientation (Tarannum et al, 2022; Stooke-Vaughan et al, 2017) making it an ideal system for the present study. Consequently, we utilized this model organism to determine whether mislocalization of importin α at the PM and subsequent spindle misorientation might lead to microcephaly or microcephaly-associated phenotypes. Leveraging our in vitro observations, we analyzed the effects of disrupting mitotic spindle orientation by abrogating importin α palmitoylation in the developing *X. laevis* model system.

*X. laevis* eggs were fertilized, placed in drug baths containing DMSO, Wnt-C59, or palmostatin at approximately NF stage 24 and allowed to develop until NF stage 42 when craniofacial measures can first be made (Fig. 4A) (Kennedy and Dickinson, 2014; Shantanam and MUELLER, 2018). Drug treated *X. laevis* embryos were assessed for

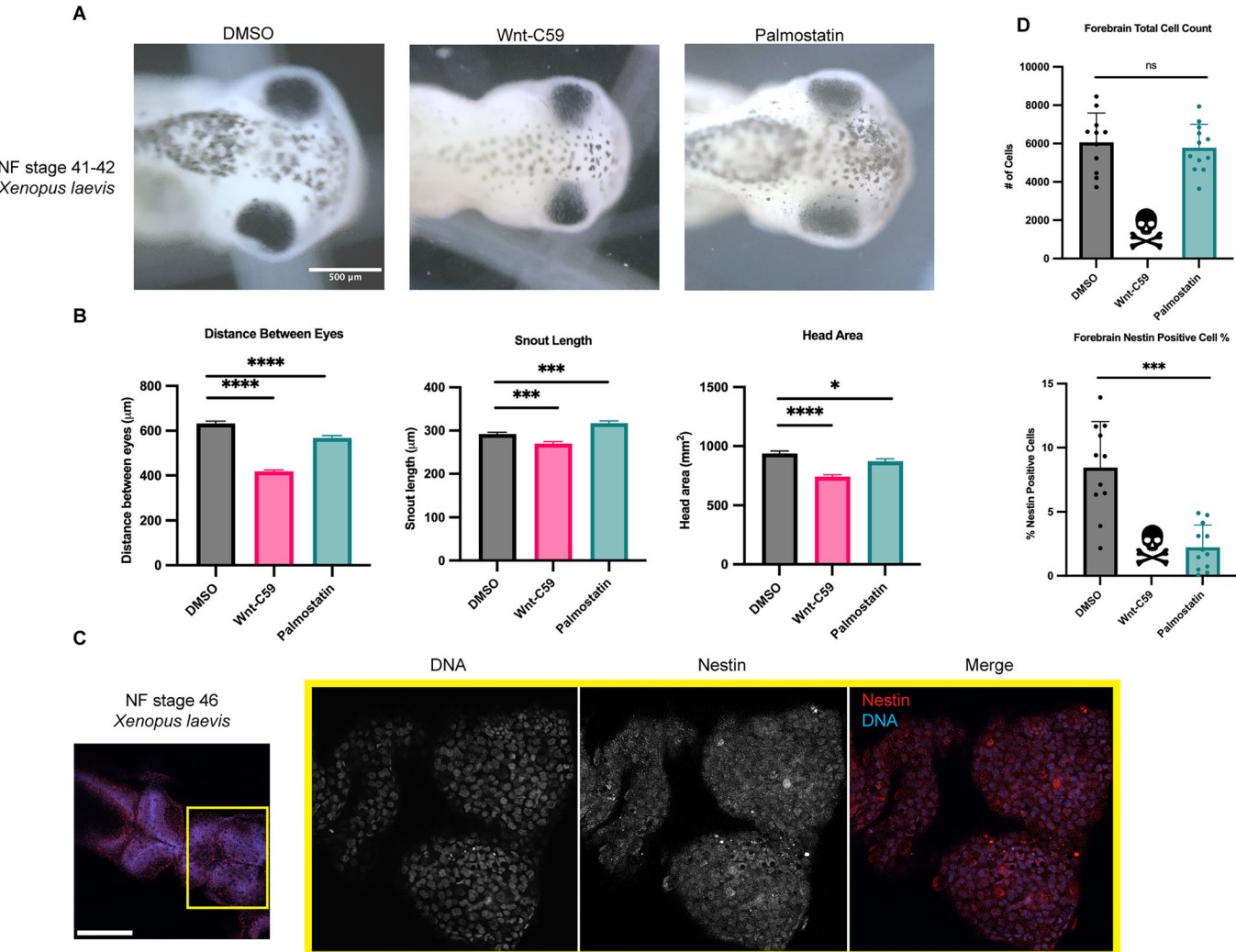

**Figure 4. Regulation of palmitoylation is required for proper brain development in *Xenopus laevis*.**

(A) Brightfield images of NF stage 42 *X. laevis* grown in the presence of DMSO, 100 μM Wnt-C59 or 1 mM palmostatin. Scale bar = 500 μm. (B) Measurements of drug treated stage 42 *X. laevis* head shape by 3 metrics: distance between eyes, snout length, and overall head area. All 3 metrics of head shape were significantly altered from DMSO control in both Wnt-C59 and palmostatin treatments. Mean +/− SEM $n > 55$. Distance between eyes ****$p < 0.0001$, snout length ***$p = 0.0003$ DMSO-Wnt-C59 $p = 0.0002$ DMSO-Palmostatin, head area *$p = 0.0389$, ****$p < 0.0001$ determined by Student's t-test. (C) Immunofluorescence images of DMSO treated NF stage 46 *X. laevis* immunostained for the neuroprogenitor marker nestin and stained with Hoechst to visualize DNA. Scale bar = 250 μm. (D) Quantification of total cell count in forebrains and percentage of forebrain cells positive for nestin signal in NF stage 46 *X. laevis* grown in the presence of DMSO, 100 μM Wnt-C59 or 1 mM palmostatin. Quantifications were performed on maximum projection images from z-stack images of *X. laevis* brains with a parent-child analysis to determine the number of total cells as determined by Hoechst signal that also were positive for nestin signal. All Wnt-C59 treated *X. laevis* embryos died before reaching NF stage 46 while all palmostatin treated *X. laevis* display a significantly reduced neuroprogenitor population by nestin positive cell count. Mean +/− SEM $n = 12$, ***$p = 0.0001$ determined by Student's t-test. Source data are available online for this figure.

craniofacial defects by three metrics: distance between eyes, snout length and overall head area. Embryos were immobilized in MS-222 and imaged from a dorsal perspective allowing for direct measurement of cranial morphometrics. Both Wnt-C59 and palmostatin treatments significantly altered all three metrics of head shape/size (Fig. 4B) indicating that precise importin α palmitoylation is required for proper craniofacial development in *X. laevis*. In addition to alterations in head shape, drug treated tadpoles also exhibited other body morphology defects. Notably, Wnt-C59 treated tadpoles exhibited much smaller tails than other conditions and a much higher mortality rate than other conditions (Fig. EV4).

While craniofacial abnormalities are a hallmark of microcephaly, we wanted to directly measure potential microcephaly-associated phenotypes. We therefore sought to quantify the neuroprogenitor population in the various drug treated *X. laevis* tadpoles. DMSO, Wnt-C59 and palmostatin treated *X. laevis* embryos were analyzed at NF stage 46, when brain development expansion reaches its apex (Exner and Willsey, 2021) by immunostaining for nestin, a neuroprogenitor marker (Suzuki et al, 2010). Immunofluorescence images were taken of tadpole forebrains and analyzed for the number of total cells, determined by DNA signal, that were positive for nestin, indicating a

neuroprogenitor identity (Fig. 4C). Due to higher mortality rate in Wnt-C59 treated *X. laevis* embryos, no Wnt-C59 treated tadpoles survived to NF stage 46 to be analyzed, indicative of severe developmental defects in both the brain and other tissue. Palmostatin treated *X. laevis* embryos survived to NF stage 46 at the same rate as DMSO controls but exhibited a significantly diminished neuroprogenitor population (Fig. 4D). While total cell count remained the same from DMSO to palmostatin treatments, the number of nestin-positive cells decreased (Fig. 4D). This decrease in neuroprogenitors at NF stage 46 suggests that the morphometric defects observed previously are indicative of true microcephaly, which is characterized by defects in neuroprogenitor population maintenance.

To further characterize the observed microcephaly-associated phenotypes NF stage 42 drug treated *X. laevis* embryos were analyzed for the number of phosphohistone 3 (PH3) positive cells, a marker for actively dividing cells (Elmaci et al, 2018), in the brain as a decrease in active divisions of cells in the brain throughout development is a hallmark of microcephaly. Embryos treated with DMSO or Wnt-C59 were whole mount immunostained for DNA and PH3 and imaged via confocal microscopy to determine the number of cells positive for PH3 in the brain (Fig. 5A). Wnt-C59 treated embryos exhibited a 6-fold decrease in the number of actively dividing cells compared to DMSO control conditions (Fig. 5D). In addition, Wnt-C59 treated embryos at NF stage 42 displayed deformed brains which lacked mid and forebrain patterning. These results together with our morphometric measurements indicate that dysregulation of palmitoylation causes craniofacial defects and particularly microcephaly in developing *X. laevis* embryos.

## Forcing importin α to the PM rescues *X. laevis* microcephaly phenotypes observed when palmitoylation is disrupted

Up to this point we have exclusively altered importin α palmitoylation using small molecule inhibitors, such as Wnt-C59 and palmostatin. Wnt-C59, however, exhibits off-target effects by inhibiting the PAT PORCN, which palmitoylates Wnt for secretion in addition to importin α. Given the crucial role of Wnt signaling in stemness and whole body development in *X. laevis* (Yu et al, 2024), it is plausible that the observed effects in Wnt-C59 treated embryos are off-target effects and cannot be directly attributed to importin α palmitoylation. Critically, to address these off-target effects we investigated whether the loss of mitotic cells in the brain leading to microcephaly was specifically caused by inhibiting importin α palmitoylation. To this end, we utilized CaaX-modified importin α to force it to the cell membrane independently of palmitoylation. Previously, we demonstrated that expression of CaaX-modified importin α could rescue spindle misorientation phenotypes induced by Wnt-C59 treatment in human cell culture (Fig. 1E). By forcing importin α to the cell membrane while importin α palmitoylation is abrogated, we can determine whether the microcephaly phenotype observed in Wnt-C59 treated *X. laevis* embryos is specifically attributable to the absence of membrane-bound importin α or other off-target effects.

We found that when *X. laevis* embryos were injected with CMV promoter-driven wild type and CaaX importin α they exhibited increased mortality and developmental defects likely due to

overexpression associated issues (Fig. EV5). To combat this, an importin α-CaaX construct was developed in a tetracycline inducible vector, pcDNA4TO, that would allow for titratable levels of importin α expression. In addition, we mitigated global developmental defects of importin α overexpression by targeting expression exclusively to the D11 blastomere of the developing embryo which is fated to give rise to the brain (Moody, 1987a, 1987b). *X. laevis* embryos were co-injected with importin α-CaaX pcDNA4TO and a plasmid expressing the tet repressor at the D11 blastomere and analyzed for potential rescue of microcephaly (Fig. 5B).

Importantly, *X. laevis* embryos microinjected with importin α-CaaX pcDNA4TO at the D11 blastomere and treated with Wnt-C59 exhibited a rescue of PH3 levels in the brain at NF stage 42 compared to uninjected Wnt-C59 treated embryos (Fig. 5D). In addition, the microinjected tadpoles exhibited brain morphology more similar to the canonical brain morphology at this stage, as seen in the DMSO control, than compared with the uninjected Wnt-C59 treated embryos (Fig. 5A,B). When treated with Wnt-C59, importin α-CaaX injected embryos showed brain PH3 levels about 2.5 times higher than both uninjected embryos and embryos injected at D11 with an mCherry tagged CaaX domain (Fig. 5C,D). Taken together, these results suggest that disrupting importin α palmitoylation leads to a loss of dividing cells in the developing *X. laevis* brain, which subsequently causes microcephaly. Furthermore, the expression of the membrane-bound importin α CaaX construct, but not the uninjected, or mCherry CaaX constructs, can rescue both brain cell proliferation and attenuate microcephaly in *X. laevis* compared to drug treatment alone. This finding aligns with our in vitro work in cell culture, where forcing importin α to the cell cortex was able to rescue spindle misorientation (Fig. 1I). These results demonstrate that the membrane localization of importin α is sufficient to rescue palmitoylation disruption-induced spindle misorientation in vitro and microcephaly in vivo.

## Discussion

Importin α is primarily recognized as a nuclear transport protein during interphase and as a spindle assembly factor in mitosis by binding NLS sequence-containing proteins. In interphase, importin α binds to NLS sequence containing proteins and transports them into the nucleus, while in mitosis importin α binds to NLS sequence-containing proteins and transports them to the developing spindles at the midline of the cell (Takeda et al, 2011; Oka and Yoneda, 2018; Goldfarb et al, 2004). In the present study, we have demonstrated a previously uncharacterized role for importin α at the PM. We have shown that importin α, when partitioned to the PM via palmitoylation, plays a crucial role in mitotic spindle orientation. Disruption of importin α palmitoylation resulted in mitotic spindle misorientation and mislocalization of the aMT anchoring protein NuMA. While LGN and Gαi are necessary to ensure proper spindle orientation in metaphase (Bergstralh et al, 2017; Neville et al, 2022; Zhong et al, 2022), we have shown that importin α, while palmitoylated, is also required for proper NuMA localization and in turn proper spindle orientation, irrespective of LGN and Gαi. Our finding that importin α plays a role in spindle orientation through interaction with the aMT anchoring complex is especially noteworthy in that importin α's ability to bind cargo is

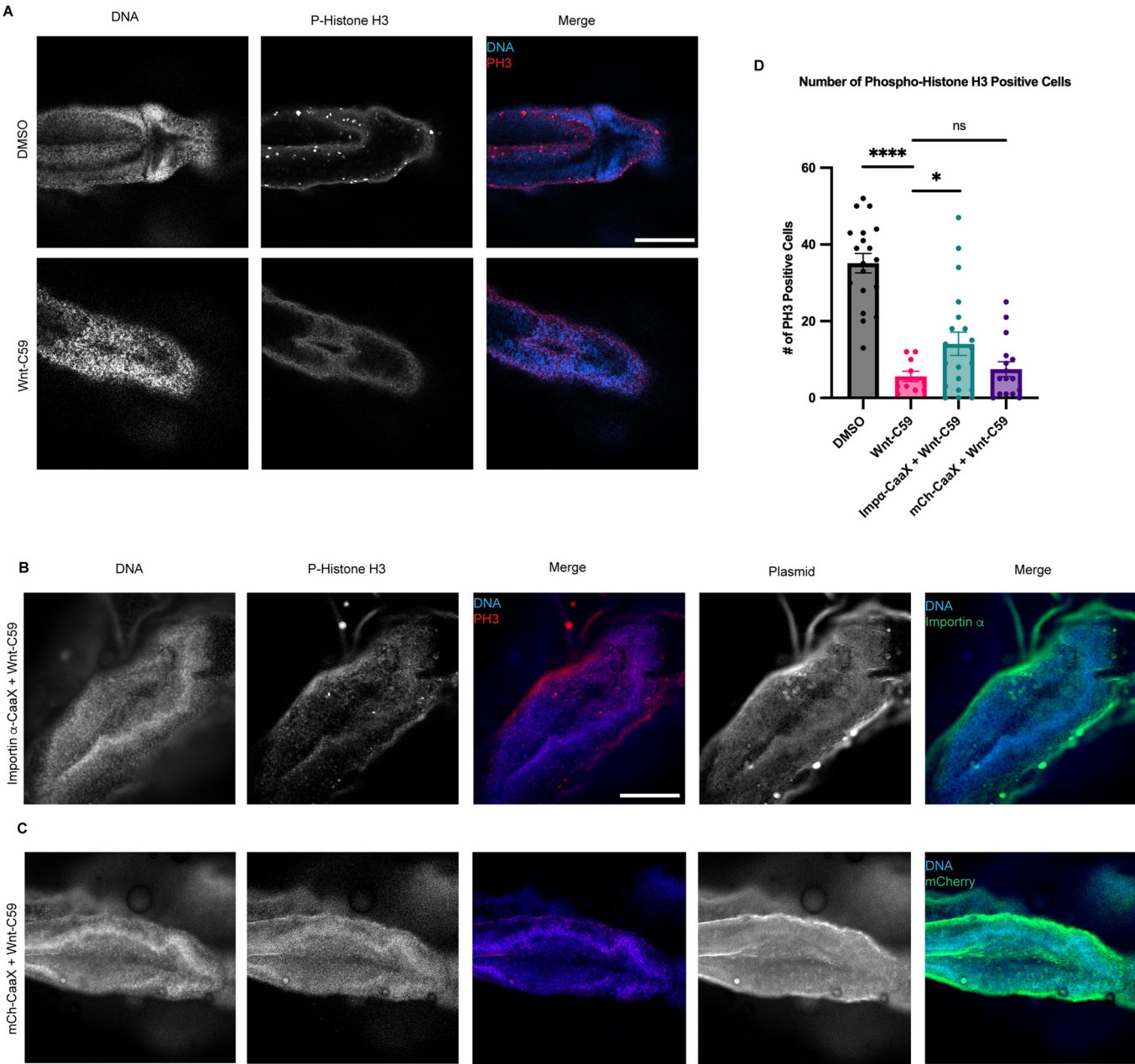

**Figure 5. Overexpression of CaaX modified importin α in the developing *X. laevis* brain partially rescues developmental defects due to PORCN inhibition.**

(A) Confocal images of NF stage 42 *X. laevis* brains from *X. laevis* grown in the presence of DMSO or 100 μM Wnt-C59 immunostained for phospho-histone H3, a marker of actively dividing cells. Scale bar = 250 μm. (B) Confocal images of NF stage 42 *X. laevis* brains from *X. laevis* expressing importin α modified with a C-terminal CaaX domain which forces cortical localization via farnesylation and grown in the presence of 100 μM Wnt-C59 immunostained for phospho-histone H3 and the modified importin α-CaaX construct. (C) Confocal images of NF stage 42 *X. laevis* brains from *X. laevis* expressing an mCherry construct modified with a C-terminal CaaX domain and grown in the presence of 100 μM Wnt-C59 immunostained for phospho-histone H3 and the modified CaaX construct. (D) Quantification of the number of phospho-histone H3 positive cells in stage 42 *X. laevis* brains of *X. laevis* grown in the presence of DMSO or 100 μM Wnt-C59 and expressing importin α-CaaX or mCherry-CaaX. Wnt-C59 treated *X. laevis* embryos showed a significantly reduced number of phospho-histone H3 positive cells in the brain compared to DMSO treated *X. laevis*. *X. laevis* embryos expressing importin α-CaaX in the brain display a partial rescue of the reduced phospho-histone H3 levels which was not recapitulated in *X. laevis* expressing mCherry-CaaX. Mean $+/-$ SEM $n > 10$, $*p < 0.0168$, $****p < 0.0001$ determined by Student's t-test. Source data are available online for this figure.

sensitive to the RanGTP gradient which emanates from the chromatin. As NuMA is a known cargo of importin α and has been shown in this work to interact with palmitoylated importin α at the polar cortex, this provides a regulatory pathway by which the localization of the chromatin and the RanGTP gradient can

determine where aMTs are anchored. This relationship is able to explain the well-defined spatial organization of the aMT anchoring complex exclusively to the polar cortex which has been shown to be essential for proper spindle orientation (Bergstralh et al, 2017) (Fig. 6).

Our results demonstrated that disruption of importin α palmitoylation leads to developmental defects in *X. laevis* embryos, specifically microcephaly. Defects in neurogenesis, in particular microcephaly, have long been linked to spindle misorientation (Taverna et al, 2014). Neuroprogenitors rely on proper spindle orientation to correctly align polarity cues which regulate cell fate determination in the developing brain (Taverna et al, 2014). During early neurogenesis, neuroprogenitors rely on several symmetric divisions to generate a large enough population of cells to later divide asymmetrically and differentiate into neurons. Spindle misorientation can cause neuroprogenitors to prematurely differentiate, depleting the pool of neuroprogenitors, resulting in an overall decrease in neuronal tissue and microcephaly (Taverna et al, 2014; Razuvaeva et al, 2023). Expression of importin α modified with a CaaX motif, which is forced to the PM by farnesylation, effectively rescued both the spindle misorientation observed in HCT116 cells and the developmental defects observed in *X. laevis* embryos. These findings further emphasize the role of palmitoylated importin α in spindle orientation and neurogenesis. The observed rescues not only demonstrate importin α's involvement in aMT anchoring, but also that the spindle misorientation and microcephaly observed when palmitoylation was disrupted by Wnt-C59 treatment are not due to off-target effects and can be attributed, in part, to mislocalization of importin α from the PM, as those phenotypes are reversed by forcing importin α to the PM.

We propose an updated model for mitotic spindle orientation in which aMT anchoring at the cell cortex is mediated by palmitoylated importin α through its interaction with NuMA (Fig. 6). In this newly proposed model, the RanGTP gradient plays a significant role in limiting the binding of NuMA to palmitoylated importin α to the polar cortex exclusively. This provides a mechanism by which palmitoylated importin α can regulate spindle orientation by binding directly to NuMA with spatial binding cues provided by the RanGTP gradient emanating from the chromosomes at the metaphase plate (Fig. 6). Notably, our newly proposed model provides a mechanistic link between factors known to affect spindle orientation, such as cell size, shape, and stretch, and the localization of aMT anchoring proteins. Given that palmitoylated importin α and its interaction with RanGTP is essential for proper localization of the aMT anchoring complex, these factors known to affect spindle orientation must exert their influence, at least in part, through palmitoylated importin α. Further studies to explore NuMA activation via phosphorylation, which has been previously linked with NuMA PM localization and association with the aMT anchoring complex of proteins (Gallini et al, 2016), and its impact on binding to palmitoylated importin α would be beneficial to expand this model and provide a robust view of NuMA localization in mitosis. It is reasonable to hypothesize that phosphorylated NuMA could bind with differential preference to palmitoylated importin α similar to our observation that hyper-palmitoylation of importin α increased NuMA/importin α interaction, but this warrants further study and remains speculative.

The present work not only highlights the importance of importin α as a key upstream regulator in mitotic spindle orientation, but also serves as the first evidence of a novel protein transport pathway by which palmitoylated importin α can transport NLS containing proteins to the PM. This new transport pathway could be involved in several cellular processes due to the abundance of NLS containing proteins that enrich at the PM and provides a

potential new level of regulation of these processes through regulation of importin α palmitoylation. Overall, our work challenges the long-standing dogma of importin α only facilitating transport into the nucleus and suggests that there are a number of potential non-canonical roles for importin α at the membrane.

# Methods

**Reagents and tools table**

| Reagent/Resource | Reference or Source | Identifier or Catalog Number |
|---|---|---|
| **Experimental models** | | |
| hTERT RPE-1 (human) | ATCC | CRL-4000 |
| HCT 116 (human) | ATCC | CRL-4000 |
| *Xenopus laevis* | NXR | N/A |
| **Recombinant DNA** | | |
| Plasmid: Importin α-mCherry-CaaX | This study | N/A |
| Plasmid: Importin α-mCherry | This study | N/A |
| Plasmid: mCherry-CaaX | This study | N/A |
| Plasmid: Importin α-mCherry-CaaX (tet inducible) | This study | N/A |
| Plasmid: Importin α-mCherry (tet inducible) | This study | N/A |
| Plasmid: mCherry-CaaX (tet inducible) | This study | N/A |
| Plasmid: pcDNA6TR | Invitrogen T-Rex system from Thermofisher | Cat # V102520 |
| Plasmid: pEGFP-C1-NuMA | Addgene | Cat # 81029 |
| Plasmid: UBC::Importin α-mCherry-HA-CaaX | This study | N/A |
| Plasmid: UBC::Importin α-mCherry-HA-CaaX | This study | N/A |
| **Antibodies** | | |
| Rabbit Polyclonal anti-NuMA | Novus Biologicals | Cat # NB100-74636 |
| Rabbit Polyclonal anti-nestin | Sino Biological | Cat # 100244-T08 |
| Monoclonal mouse living colors antibody | Takarabio | Cat # 632460 |
| Rabbit polyclonal anti-mCherry | Proteintech | Cat # 26765-1-AP |
| Monoclonal mouse anti-importin α | Proteintech | Cat # 66870-1-Ig |
| Rabbit polyclonal anti-importin α | ABclonal | Cat # A1623 |
| Monoclonal mouse anti-β tubulin | DSHB | Antibody E7 |
| Monoclonal mouse anti-phospho histone H3 | Proteintech | Cat # 66863-1-Ig |
| Monoclonal mouse anti-SAP97 | Enzo Life Sciences | Cat # 06021547 |
| Rabbit polyclonal anti-γ tubulin | Sigma-Aldrich | Cat # T5192 |

| Reagent/Resource | Reference or Source | Identifier or Catalog Number |
|---|---|---|
| Monoclonal mouse anti-HA | Sigma-Aldrich | Cat # SAB2702196 |
| Donkey anti-mouse IgG-AF488 | Southern Biotech | Cat # 6411-30 |
| Donkey anti-rabbit IgG-AF488 | Southern Biotech | Cat # 6440-30 |
| Donkey anti-mouse IgG-AF568 | Invitrogen | Cat # A10037 |
| Donkey anti-rabbit IgG-AF568 | Invitrogen | Cat # A10042 |
| **Oligonucleotides and other sequence-based reagents** | | |
| **Chemicals, Enzymes and other reagents** | | |
| Palmostatin-B | Sigma-Aldrich | Cat # 178501-5MG |
| Wnt-C59 | Selleck Chem | Cat # S7037 |
| Importazole | Selleck Chem | Cat # S8446 |
| Ivermectin | ThermoFisher | Cat # J6277.03 |
| 2-BromoPalmitate | Sigma-Aldrich | Cat # 21604-1G |
| MG-132 | Abcam | Cat # Ab141003 |
| RO-3306 | VWR International | Cat # 102516-266 |
| McCoy's 5A Media | Avantor Scientific | Cat # 45000-374 |
| Fetal bovine serum (FBS) | Avantor Scientific | Cat # 76327-086 |
| PBS | Avantor Scientific | Cat # 45000-446 |
| Gentamycin | Avantor Scientific | Cat # 102614-744 |
| Duolink® In Situ Detection Reagents Red | Sigma-Aldrich | Cat # DUO92008 |
| SE cell line 4D nucleofector X kit S | Lonza Bioscience | Cat # V4XC-1032 |
| **Software** | | |
| Celleste image analysis software version 6.0 | Invitrogen | Cat # AMEP4942 |
| **Other** | | |
| Lonza 4D Nucleofector X-unit | Lonza Bioscience | Cat # AAF-1003X |

## Methods and protocols

### Cell culture

RPE-1 and HCT116 cells were cultured as previously described (Kiyomitsu and Cheeseman, 2012). Cell cultures were tested for mycoplasma contamination every 6 months.

## Animal models

Adult Female XLA.NXR-WTNXR (NXR_0031) frogs were purchased from the National Xenopus Resource and maintained at the Stony Brook University animal facility. Frogs were maintained in a pathogen-free facility held at 18 °C, housed socially in aquatic tanks under 12:12 h light:dark cycles and fed three times weekly with a pelleted Xenopus diet. All experimental procedures were approved by the Institutional Animal Care and Use Committee. All animals were maintained in accordance with standards established by the Division of Laboratory Animal Resources at Stony Brook University.

## Cell culture and immunostaining

RPE-1 and HCT116 cells were cultured as previously described (Kiyomitsu and Cheeseman, 2012) in DMEM F-12 and McCoy's 5A media, respectively, supplemented with 5% FBS and grown at 5% $CO_2$. Immunostaining was carried out on cells cultured onto fibronectin-coated coverslips, fixed with 4% PFA, permeabilized with PBS + 0.2% Triton X-100 (Sigma-Aldrich 9036-19-5), and blocked with Bovine Serum Albumin (BSA) in PBS + 0.2% Triton X-100. The coverslips were than incubated with antibodies diluted in PBS + 0.2% Triton X-100 as follows: Rabbit polyclonal anti-NuMA 1:1000 (Novus Biologicals), monoclonal living colors antibody 1:1000 (Takarabio), rabbit polyclonal anti-mcherry 1:1000 (Proteintech), monoclonal anti-importin α 1:1000 (Proteintech), rabbit polyclonal anti-importin α 1:1000 (ABclonal), monoclonal anti-β tubulin E7 1:1000 (DSHB), monoclonal anti-SAP97 1:1000 (Enzo Life Sciences), rabbit polyclonal anti-γ tubulin 1:1000 (Sigma-Aldrich), donkey anti-mouse IgG AF-488 1:1000 (Southern Biotech), donkey anti-rabbit IgG AF-488 1:1000 (Southern Biotech), donkey anti-mouse IgG AF-568 1:1000 (Invitrogen) and donkey anti-rabbit IgG AF-488 1:1000 (Invitrogen). The coverslips were then mounted onto slides with ProLong Diamond Antifade Mountant (ThermoFisher P36961).

## Mitotic arrest and drug treatment

RPE-1 and HCT116 cells were arrested in metaphase through a sequential drug treatment of RO-3306 and MG-132. Cells were treated with 9 μM RO-3306 to arrest at the G2/M transition for 20 h at 37 °C. Cells were then washed with fresh media three times to remove RO-3306 and treated with 20 μM MG-132 within 15 min of RO-3306 washout to arrest cells at metaphase. Cells were incubated in MG-132 for 1 h at 37 °C. In experiments with palmitoylation and importin function disrupting drug treatments cells were treated with DMSO, 10 μM Wnt-C59, 50 μM palmostatin, 40 μM importazole, 25 μM ivermectin, or 100 μM 2-bromopalmitate with the MG-132 treatment and incubated for 1 h at 37 °C.

## Spindle angle measurement

Mitotic spindle angles of metaphase-arrested cells were determined by measuring the vertical and horizontal distances between the centrosomes at each pole of the mitotic cell and calculating the arctangent of the vertical distance divided by the horizontal distance as follows:

$$\alpha = \arctan\left(\frac{z}{x}\right)$$

The vertical distance was determined through imaging the cell and measuring the length of a line drawn between each centrosome. The horizontal distance was determined through using a z-stack of

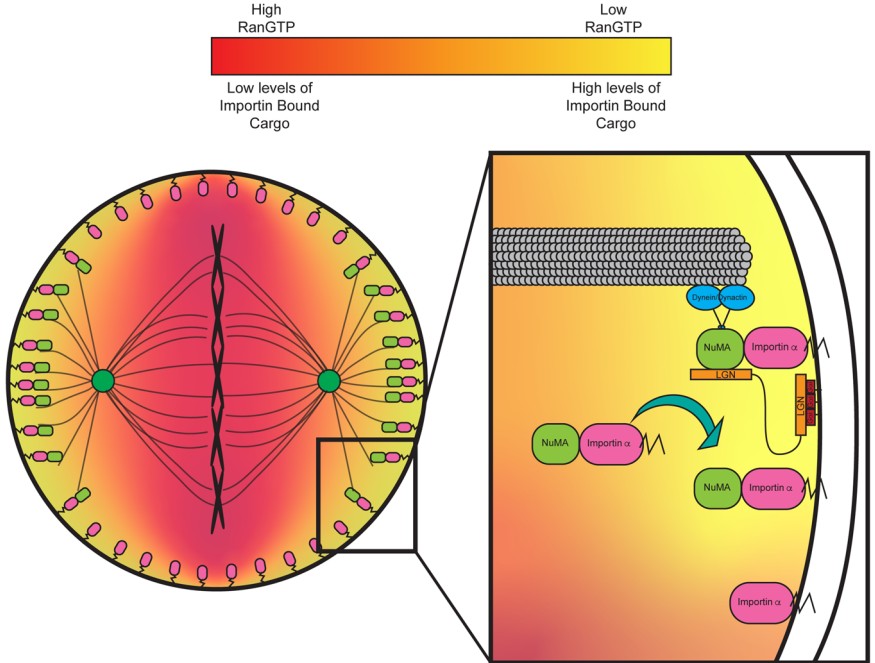

**Figure 6.  Importin α regulates mitotic spindle orientation through mediating NuMA localization to the metaphase cortex and maintenance at the cell cortex through anaphase in a palmitoylation-dependent manner.**

Proposed model of palmitoylated importin α's role in astral microtubule anchoring as a transporter of NuMA and a scaffold at the cell cortex for astral microtubule anchoring proteins to maintain cortical localization throughout metaphase and anaphase.

the mitotic cell, determining at which z-slice each centrosome was in optimal focus, and calculating the z distance between each slice of optimal focus. Z-stacks were taken at a step distance of 0.1 μm with varying numbers of steps depending on individual cell size.

## *X. laevis* fertilization

*X. laevis* adult females were induced to lay eggs by a priming injection of 100 U of pregnant mare serum gonadotropin (PMSG) at least 48 h before use and a boosting injection of 500 U of human chorionic gonadotropin (hCG) 16 h before use. Following the hCG injection, adult female *X. laevis* were placed in a 2 L water bath of 1X MMR (100 mM NaCl, 2 mM KCl, 1 mM MgCl$_2$, 2 mM CaCl$_2$, 0.1 mM EDTA, 5 mM HEPES pH 7.8) overnight at 17 °C. Approximately 16 h following hCG injection, fresh eggs were collected by squeezing eggs from ovulating frogs into a 10 cm plastic petri dish. To fertilize eggs, a sperm solution made from ¼ of a male frog testis was placed in 1 mL 1X MR (100 mM NaCl, 1.8 mM KCl, 2.0 mM CaCl$_2$, 1.0 MgCl$_2$, 5.0 mM HEPES-NaOH, pH 7.6) and homogenized using scissors and a pestle. 1 mL of sperm solution was added dropwise to the freshly squeezed eggs and the dish was swirled to form a monolayer of eggs and incubated for 3 min. Dishes were flooded with milli-Q water and incubated for an additional 10 min. Eggs were dejellied with a 2% cysteine solution for 6 min with occasional swirling and washed 5 times with 1/3 MR. Fertilized eggs were incubated at 23 °C until the appropriate developmental stage. In experiments where *X. laevis* embryos were drug treated, the embryos were placed into a bath of 1/3 MR containing either DMSO, 100 μM Wnt-C59, or 1 mM palmostatin

24 h post fertilization and kept in drug bath until the appropriate developmental stage for each experiment. In drug treatment conditions, morphometric defects made it often difficult to determine the exact stage of Wnt-C59 and palmostatin treated embryos between NF stages 40 to 43, so analysis was conducted when the DMSO treated embryos were at the appropriate stage.

## *X. laevis* morphometric measurements

*X. laevis* embryos were analyzed for morphometric defects at NF stage 42 by immobilizing embryos in a bath of 140 μg/mL MS-222 and imaging at 4X mounted upright. All measurements were made in ImageJ. Distance between eyes was determined by measuring the straight line distance from the right most portion of the left eye to the left most portion of the right eye, snout length was determined by drawing a line from the front of the left eye to the right eye and then measuring the straight line distance from the center of this line to the mouth, and overall head area was determined by measuring the area of a circle drawn around the head such that each eye is completely within the circle and the circle does not extend beyond the snout.

## *X. laevis* whole mount immunostaining

*X. laevis* embryos were fixed in 4% PFA for 24 h at 4 °C. Following fixation, embryos were immunostained by washing in PBS 3 × 20 min, photobleaching in a solution of 5% formamide and 1.2% hydrogen peroxide for 2 h, washing in PBS + 0.1% Triton X-100 (PBST) overnight at 4 °C, blocking with 2% BSA in PBST for

3 h at RT, incubating with 1° antibodies diluted in PBST overnight at 4 °C as follows: rabbit polyclonal anti-nestin 1:1000 (Sino Biological), rabbit polyclonal anti-mCherry 1:1000 (Proteintech), and monoclonal anti-phospho Histone H3 1:1000 (Proteintech), washing $3 \times 1$ h in PBST, incubating with 2° antibodies diluted in PBST overnight at 4 °C as follows: washing $3 \times 1$ h in PBST and mounting onto a coverslip in fluoromount G. In cases where embryos were cleared before mounting, embryos were chilled in 1-propanol and incubated $2 \times 5$ min, cleared with 5 mL Murray's (2 parts Benzyl Benzoate and 1 part Benzyl Alcohol) and then mounted onto a coverslip with fluoromount G.

## Plasmid construct development

Importin α-mCherry plasmid constructs were cloned into pCS2+ and pcDNA4TO vectors from existing plasmid constructs of importin α, GFP, mCherry-CaaX, and NuMA-GFP. Importin α-mCherry-HA plasmid constructs were designed with vectorbuilder using UBC promoter plasmids.

## Embryo microinjection

Plasmid was loaded into a needle pulled from a 1 mm glass capillary tube (TW100F-3, World Percision Instruments) using a L/M-3P-A electrode/needle puller. Embryos were placed in a mesh-bottomed plastic dish with 2.5% Ficoll in 1/3 MR and microinjected with a 2nL droplet of the appropriate plasmid using a Narishige IM-400 microinjector system equipped with a MM-3 micromanipulator (Narishige). For stage 1 injections embryos were injected directly at the animal pole, for stage 2 injections 1 blastomere was injected at the animal pole, and for stage 5 injections the D11 blastomere was injected at roughly the middle of the blastomere (as per Moody 1987a). pCS2+ plasmids were injected at a concentration of 10 ng/μL such that the final concentration of plasmid delivered was 20 pg. pcDNA4TO plasmids were co-injected with pcDNA6TR at concentrations of 5 ng/μL and 25 ng/μL, respectively, such that the final concentration of total plasmid delivered was 60 pg. Following injection, embryos were placed into a new dish containing 2.5% Ficoll in 1/3 MR and incubated at 23 °C for 4 h after which embryos were moved to a dish containing 1/3 MR and incubated at 23 °C until appropriate developmental stage. Embryos injected with pcDNA4TO + pcDNA6TR were placed in 1/3 MR containing 12.5 μg/mL doxycycline to induce gene expression 4 h post injection and transferred to fresh 1/3 MR with 12.5 μg/mL doxycycline 24 h post injection.

## X. laevis nestin positive cell count

NF Stage 46 X. laevis embryos were whole mount fixed and stained for DNA (Hoechst) and nestin (Rb α-nestin Sino Biological 100244-T08). Embryos were imaged at 20X magnification on an EVOS M7000 epifluorescent microscope to generate a z-stack image of the brain of each embryo. Z-stack images were processed in Celeste Image Analysis software for 3D-deconvolution to remove background signal and using a parent child analysis measured the number of DNA containing cells in the brain with overlapping nestin signal in the maximum projection of the processed z-stack.

## X. laevis phospho-histone H3 positive cell count

NF Stage 42 X. laevis embryos were whole mount fixed and stained for DNA (Hoechst), Phospho-histone H3, and mCherry to visualize mCherry tagged proteins from microinjected constructs. Whole embryos were imaged at 10X on a Zeiss LSM 980 confocal microscope by imaging a Z-stack of the brain. A maximum projection image from this z-stack was then processed in Celeste Image Analysis Software and a 3D count was measured in an ROI around the brain to determine the number of cells positive for Phospho-histone H3 signal. Phospho-histone H3 positive cell counts were determined by thresholding the minimal signal such that individual positive cells could be resolved from background fluorescence.

## Proximity ligation assay

Proximity ligation assay (PLA) was performed using DuoLink PLA (Millipore Sigma) following the recommended protocol. PLA analysis of importin α and NuMA interaction was performed in mitotically arrested HCT 116 cells using mouse anti-importin α (Proteintech) and rabbit anti-NuMA (Novus Biologicals). PLA analysis of importin α and Dlg interaction was performed in mitotically arrested HCT 116 cells using rabbit anti-importin α (ABclonal) and mouse anti-SAP97 (Enzo Life Sciences). Cells were imaged with an EVOS M7000 epifluorescent microscope at 60X magnification. Localization of PLA fluorescent signal was quantified by counting the number of foci within three separate ROIs of each cell. Polar cortex ROI was defined as the region from the plasma membrane at each pole to the centrosomes. Lateral cortex ROI was defined as the region from the plasma membrane to the chromatin between the centrosomes. Cytosol ROI was defined as the region between the centrosomes excluding the plasma membrane. The number of foci in each region was calculated as a percentage of the total number of foci for that cell. Cell border was determined by a brightfield image of each mitotic cell quantified.

## Cell lysis and western blot analysis

Lysates of RPE-1 and HCT 116 cells were generated from 10 cm dishes seeded with $1 \times 10^6$ cells 2 days prior to lysis. Cells were collected by washing with ice-cold PBS and scraping off the plate into solution. Cells were spun at $100 \times g$ for 5 min, supernatant was aspirated and cells were resuspended in 150 μL RIPA buffer (150 mM sodium chloride, 1.0% Triton X-100, 0.5% sodium deoxycholate, 0.1% SDS, 50 mM Tris, pH 8.0) supplemented with 10 μg/mL each of leupeptin, pepstatin and chymostatin (LPC) protease inhibitors. Resuspended cells were rocked at 4 °C for 1 h and spun at 12,000 rpm for 20 min at 4 °C in an Eppendorf FA-45-24-11 rotor. Supernatant containing cell lysate proteins was then mixed 1:1 with 2X laemlli buffer, boiled at 100 °C for 5 min and stored at −20 °C until use. Western blot analysis was performed on cell lysates by running lysates through SDS-PAGE in a 7.5% or 5.0% Tris-glycine gel (dependent on size of proteins being analyzed), transferring to a nitrocellulose membrane, and blotting for target proteins. Western blot analysis for NuMA was performed with overnight transfer of SDS-PAGE gel at 20 V at 4 °C while all

other proteins were performed with a transfer at 150 V for 90 min at room temperature.

## Co-immunoprecipitation

Co-immunoprecipitation was performed with Thermo Fisher IgG conjugated magnetic Dynabeads following recommended protocol. Cell lysates were generated for immunoprecipitation experiments with previously stated cell lysis protocol using a non-denaturing lysis buffer (20 mM Tris HCl pH 8, 137 mM NaCl, 1% Triton X-100, 2 mM EDTA) in place of RIPA buffer.

## Mitotic protein localization measurement

To determine changes to importin α (KPNA2) and NuMA localization in cultured cells upon drug treatment, cells were mounted onto fibronectin-coated coverslips, arrested in metaphase, drug treated, washed with cytoskeletal buffer (100 mM NaCl, 300 mM Sucrose, 3 mM MgCl$_2$, 10 mM PIPES, pH 6.9, supplemented with 250 μL 1 M EGTA and 250 μL Triton X-100 per 50 mL immediately before use), fixed with 4% PFA, and immunostained for DNA and KPNA2/NuMA. For KPNA2 localization cells were imaged using an EVOS M7000 at 100X with cell boundaries determined using bright-field images. For NuMA localization cells were imaged using a Leica SP5 confocal at 40X. To determine the cellular localization of target protein in each drug condition, 60 mitotic cells were imaged per drug treatment and the fluorescent intensity of KPNA2 or NuMA signal was measured in ImageJ at three cellular locations. A 10 pixel wide and 50 pixel long line was drawn at one cortical pole, one lateral membrane, and along the midline of the cell and measured. In order to normalize variations in intensity from inconsistent immunostaining, these measurements were normalized to each other on a cell-by-cell basis by determining the ratio of polar vs lateral signal, polar vs cytosolic signal, and lateral vs cytosolic signal. In the cases where one cortical pole differed in intensity from the opposite cortical pole, the pole with the higher measure of intensity was used for data analysis (the same method was used when measuring the lateral poles).

## DNA transfection

HCT116 cells were seeded onto fibronectin-coated coverslips. The following day media was replaced with serum-free media and a mixture of 1 μg plasmid in 12 μL polyethylenimine (PEI) was added dropwise to cells. Cells were incubated in PEI mixture for 4 h then media was washed out and replaced with complete media and incubated overnight before fixation and imaging.

## Subcellular fractionation

HCT116 cells were seeded in a 10 cm dish and incubated at 37 °C. Following 2 days of incubation, cells were incubated with DMSO for 1 h at 37 °C then lifted from dish with ice-cold PBS and a cell scraper. Collected cells were then fractionated using the Minute Plasma Membrane/Protein Isolation and Cell Fractionation kit from Invent Biotechnologies following the recommended protocol.

## DNA nucleofection

HCT116 cells were transfected via nucleofection using LONZA SE cell line 4D-Nucleofector kit (Catalog #V4XC-1032) following recommended protocol. Following nucleofection cells were incubated for 24 h before fixation and immunostaining for mitotic spindle angle analysis.

## Palmitoylation prediction

GPS-Palm (Ning et al, 2021) was employed for detection of potential palmitoylated cysteines within human KPNA2. Cysteines above a threshold score of >0.6 (specificity >85% and accuracy >82%) were considered to be likely palmitoylated.

## Nuclear localization signal, cellular localization, and protein function prediction

NucPred (Brameier et al, 2007) was used to determine which proteins in the human genome contain potential NLS sequences. Any proteins above a threshold score of >0.63 (specificity >71% and accuracy >53%) were considered potential NLS-sequence containing candidate proteins. Proteins were then filtered to discard transmembrane proteins while retaining only plasma membrane proteins, as identified by UniProt GO identifiers (Ashburner et al, 2000; Aleksander et al, 2023). Proteins were then sorted by cellular localization and function using GO enrichment analysis (Thomas et al, 2022).

## Quantification and statistical analysis

All statistical analysis was performed in GraphPad Prism 10.0. Comparisons between datasets was determined by a student's t-test unless otherwise stated. Graphs represent the mean value +/− the SEM unless otherwise stated. $*p < 0.05$, $**p < 0.01$, $***p < 0.001$, $****p < 0.0001$ unless otherwise stated. Blinding was not performed unless otherwise stated.

## Schematic image generation

Schematics for main text of manuscript were developed in adobe illustrator. Synopsis schematic was developed in BioRender:
Created in BioRender. https://BioRender.com/b0co5xm.

# Data availability

The microscopy data from this publication has been deposited to the BioImage Archive database: https://www.ebi.ac.uk/biostudies/bioimages/studies/S-BIAD1801 and assigned the identifier: https://doi.org/10.6019/S-BIAD1801.

The source data of this paper are collected in the following database record: biostudies:S-SCDT-10_1038-S44319-025-00484-8.

# Peer review information

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

## Acknowledgements

We thank Natalie Mosqueda, Kathryn Malone, and Melanie Garcia for their constructive feedback on this work, engaging discussions, and edits during the writing of this manuscript. We thank Maurice Kernan, Holly Colognato, Gerald

Thomsen, and Daniel Levy for their insights and suggestions on experimental approach. We thank the Stony Brook University Central Microscopy Imaging Center (CMIC) and Guowei Tian for assistance in imaging with Zeiss LSM 980 Airsycan 2 NLO Two-Photon Confocal Microscope. Work was supported by National Institutes of Health grant 1R35GM147569-01 (CWB).

## Author contributions

**Patrick James Sutton**: Data curation; Formal analysis; Investigation; Methodology; Writing—original draft; Writing—review and editing. **Natalie Mosqueda**: Investigation; Writing—review and editing. **Christopher W Brownlee**: Conceptualization; Resources; Supervision; Funding acquisition; Project administration; Writing—review and editing.

Source data underlying figure panels in this paper may have individual authorship assigned. Where available, figure panel/source data authorship is listed in the following database record: biostudies:S-SCDT-10_1038-S44319-025-00484-8.

## Disclosure and competing interests statement

The authors declare no competing interests.

# Expanded View Figures

**A**

| Position | Peptide | Score |
|---|---|---|
| 133 | KFVSFLGRTD**C**SPIQFESAWA | 0.288 |
| 223 | LAVPDMSSLA**C**GYLRNLTWTL | 0.6991 |
| 237 | RNLTWTLSNL**C**RNKNPAPPID | 0.6514 |
| 272 | HDDPEVLADT**C**WAISYLTDGP | 0.1746 |
| 419 | TVEQIVYLVH**C**GIIEPLMNLL | 0.1964 |
| 467 | TEKLSIMIEE**C**GGLDKIEALQ | 0.7908 |

**B**

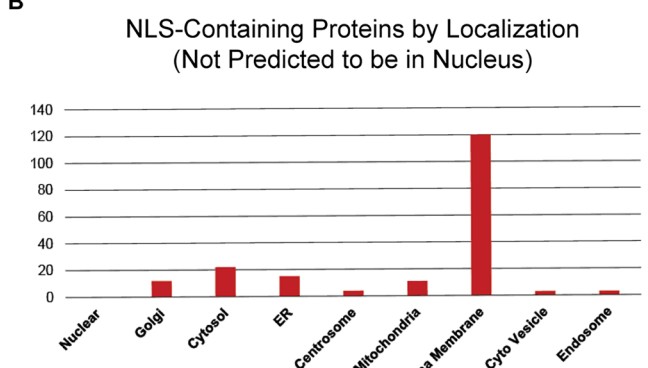

**C**

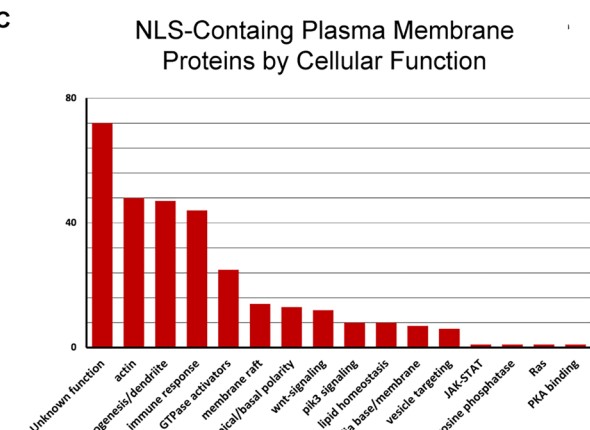

**Figure EV1.  Proteome screens confirm palmitoylation of human importin α-1 and enrichment of NLS containing proteins by cellular localization and function.**

(A) Prediction of palmitoylated cysteine residues in Human Importin α-1 (KPNA2) by GPS-Palm. Prediction score is on a 0-1 scale with 0 representing low confidence of palmitoylation and 1 representing high confidence of palmitoylation. Three highest confidence residues are highlighted in green. (B) Cellular localization of NLS containing proteins not predicted to be in the nucleus. (C) Cellular functions of NLS containing proteins found at the PM sorted by gene ontology terms.

**A**

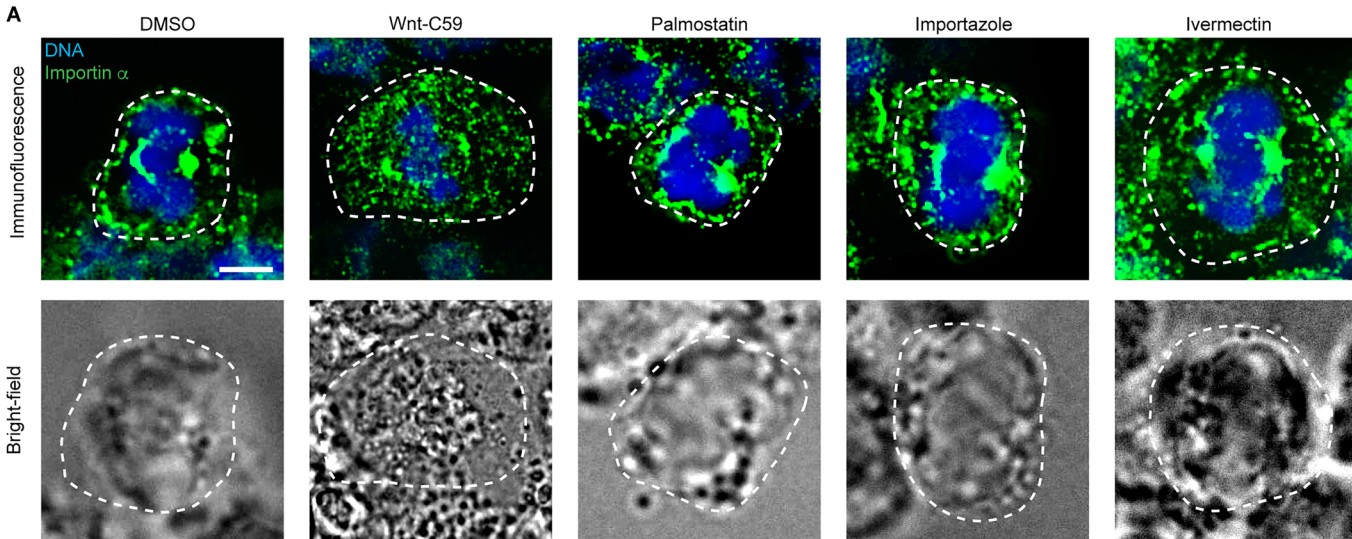

**B**

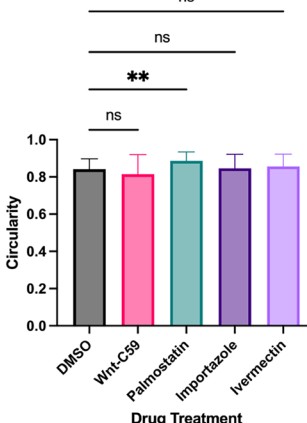

Figure EV2.   KPNA2 cellular localization boundary determination.

(**A**) Representative immunofluorescence and bright-field images for cells used in Fig. 1A to determine KPNA2 localization and cell boundary determination for each drug treatment. Scale bar = 5 μm. (**B**) Quantification of the circularity of metaphase-arrested HCT116 cells treated with DMSO, 10 μM Wnt-C59, 50 μM palmostatin, 40 μM importazole or 25 μM ivermectin for 1 h. Mean +/− SEM $n = 60$, \*\*$p = 0.0031$ determined by Student's t-test.

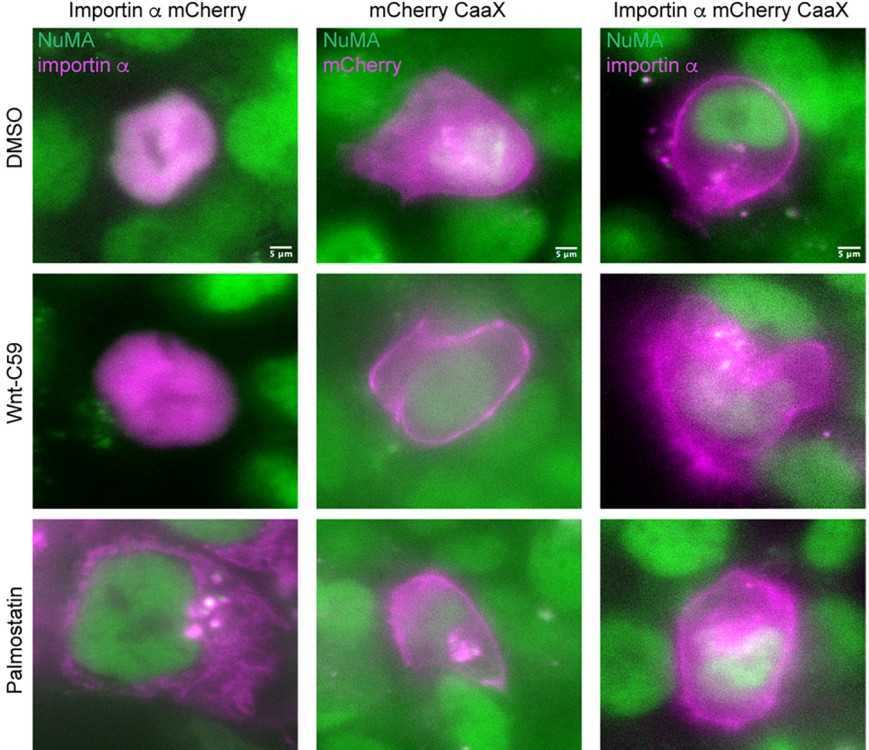

**Figure EV3. CaaX modified importin α localizes to the plasma membrane independent of palmitoylation.**

Immunofluorescent images of HCT116 cells transfected with importin α-mCherry, mCherry-CaaX or importin α-mCherry-CaaX treated with DMSO, Wnt-C59 or palmostatin. Importin α-mCherry-CaaX localizes to the plasma membrane in all drug treatments. Scale bar = 5 µm.

**A**    DMSO treated NF stage 42 *X. laevis*

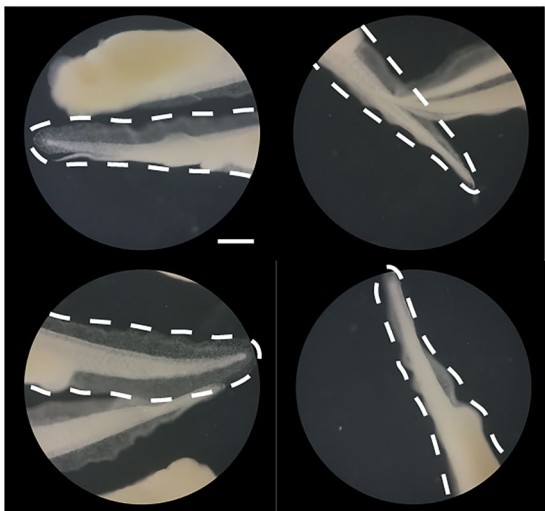

**B**    Wnt-C59 treated NF stage 42 *X. laevis*

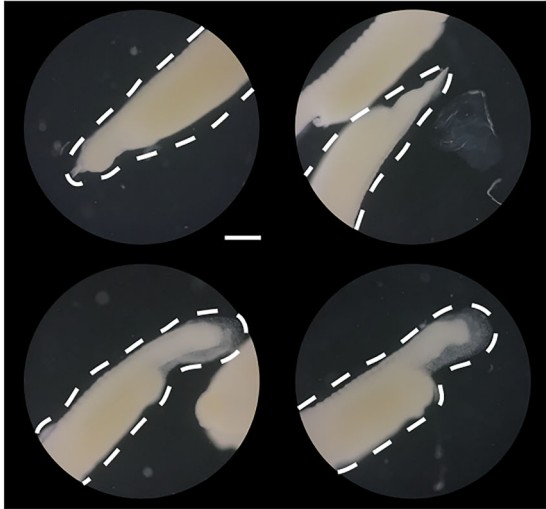

**Figure EV4.   Wnt-C59 treated *X. laevis* tadpoles exhibit shortened tail length.**

(A) Brightfield images of NF Stage 42 *X. laevis* tadpole tails treated with DMSO from 24 to 72 hpf. DMSO treated tadpoles exhibit normal development and typical length tails. Scale bar = 500 μm. White dashed line indicates examined tadpole tail for each image. (B) Brightfield images of NF Stage 42 *X. laevis* tadpole tails treated with Wnt-C59 from 24 to 72 hpf. Wnt-C59 treated tadpoles exhibit abnormal development, shortened tails, and abnormally shaped tails. Scale bar = 500 μm. White dashed line indicates examined tadpole tail for each image.

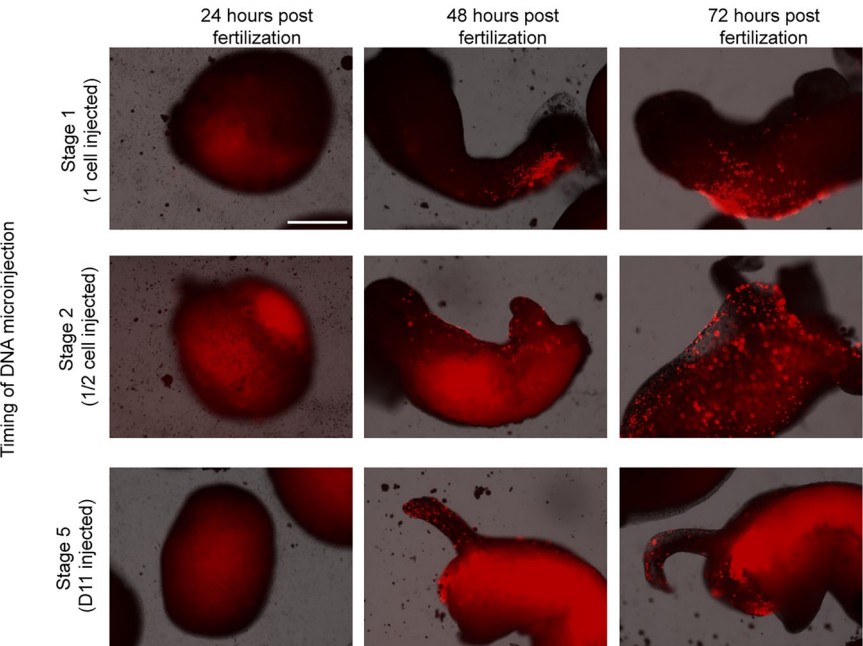

**Figure EV5.  Importin α overexpression produces severe developmental defects in *X. laevis* embryos.**

Immunofluorescent images of *X. laevis* embryos co-injected with importin α-mCherry-CaaX pcDNA4TO and pcDNA6TR at 24, 48, and 72 h post fertilization. Embryos were injected at either the 1 cell, 2 cell (injected into 1 of 2 cells), or 16 cell stage (injected into the D11 blastomere). All 1 and 2 cell injected embryos exhibit high mortality and severe developmental defects. D11 injected embryos had better survivability though some embryos still exhibited defects. Scale bar = 500 µm.

