## [Peer Review File · EMBO Reports]

Palmitoylated Importin α Regulates Mitotic Spindle Orientation Through Interaction with NuMA

Patrick Sutton, Natalie Mosqueda, and Christopher Brownlee

Corresponding author(s): Christopher Brownlee (Christopher.Brownlee@stonybrook.edu) , Patrick Sutton (patrick.sutton@stonybrook.edu)

Review Timeline:

Transfer Date:	31st Oct 24
Editorial Decision:	7th Nov 24
Revision Received:	6th Feb 25
Editorial Decision:	21st Mar 25
Revision Received:	2nd Apr 25
Accepted:	7th May 25

Editor: Deniz Senyilmaz Tiebe

Transaction Report: This manuscript was transferred to EMBO reports following peer review at The EMBO Journal.

Dear Dr. Brownlee,

Thank you for transferring your manuscript to EMBO Reports, which was previously reviewed at another venue.

Having read the manuscript, referee reports and your revision plan, I would like to invite you to submit a revised manuscript to EMBO Reports, as previously communicated. The revised manuscript will be reevaluated by referees #2 and #3, and an additional expert may be involved if need be.

Please address all referee concerns in a complete point-by-point response. Acceptance of the manuscript will depend on a positive outcome of a second round of review. It is EMBO reports policy to allow a single round of major experimental revision only and acceptance or rejection of the manuscript will therefore depend on the completeness of your responses included in the next, final version of the manuscript.

We realize that it is difficult to revise to a specific deadline. In the interest of protecting the conceptual advance provided by the work, we recommend a revision within 3 months. Please discuss the revision progress ahead of this time with me if you require more time to complete the revisions, or if you have questions or comments regarding the revision (also by video chat).

1. A data availability section providing access to data deposited in public databases is missing (where applicable).
2. Your manuscript contains statistics and error bars based on $n=2$. Please use scatter plots in these cases.

You can submit the revision either as a Scientific Report or as a Research Article. For Scientific Reports, the revised manuscript can contain up to 5 main figures and 5 Expanded View figures, and it should not exceed 27000 characters. If the revision leads to a manuscript with more than 5 main figures it will be published as a Research Article. In this case the Results and Discussion section should be separate. If a Scientific Report is submitted, these sections have to be combined. This will help to shorten the manuscript text by eliminating some redundancy that is inevitable when discussing the same experiments twice. In either case, all materials and methods should be included in the main manuscript file.

4) a .docx formatted letter INCLUDING the reviewers' reports and your detailed point-by-point responses to their comments. As part of the EMBO publication's Transparent Editorial Process, EMBO reports publishes online a Review Process File (RPF) to accompany accepted manuscripts. This File will be published in conjunction with your paper and will include the referee reports, your point-by-point response and all pertinent correspondence relating to the manuscript.

<https://www.embopress.org/page/journal/14693178/authorguide#transparentprocess>

5) a complete author checklist, which you can download from our author guidelines <https://www.embopress.org/page/journal/14693178/authorguide>. Please insert information in the checklist that is also reflected in the manuscript. The completed author checklist will also be part of the RPF.

6) Please note that all corresponding authors are required to supply an ORCID ID for their name upon submission of a revised manuscript (<<https://orcid.org/>>). Please find instructions on how to link your ORCID ID to your account in our manuscript tracking system in our Author guidelines <<https://www.embopress.org/page/journal/14693178/authorguide#authorshipguidelines>>

Additional information on source data and instruction on how to label the files are available: <https://www.embopress.org/page/journal/14693178/authorguide#sourcedata>

9) Our journal encourages inclusion of *data citations in the reference list* to directly cite datasets that were re-used and obtained from public databases. Data citations in the article text are distinct from normal bibliographical citations and should directly link to the database records from which the data can be accessed. In the main text, data citations are formatted as follows: "Data ref: Smith et al, 2001" or "Data ref: NCBI Sequence Read Archive PRJNA342805, 2017". In the Reference list, data citations must be labeled with "[DATASET]". A data reference must provide the database name, accession number/identifiers and a resolvable link to the landing page from which the data can be accessed at the end of the reference. Further instructions are available at <http://www.embopress.org/page/journal/14693178/authorguide#referencesformat>

12) Please also note our reference format: <http://www.embopress.org/page/journal/14693178/authorguide#referencesformat>

13) All Materials and Methods need to be described in the main text using our 'Structured Methods' format, which is required for

all research articles. According to this format, the Methods section includes a Reagents and Tools Table (listing key reagents, experimental models, software and relevant equipment and including their sources and relevant identifiers) followed by a Methods and Protocols section describing the methods using a step-by-step protocol format. The aim is to facilitate adoption of the methodologies across labs. More information on how to adhere to this format as well as a downloadable template (.docx) for the Reagents and Tools Table can be found in our author guidelines:
<https://www.embopress.org/page/journal/14693178/authorguide#structuredmethods>.

An example of a Method paper with Structured Methods can be found here:
<https://www.embopress.org/doi/10.15252/msb.20178071>.

I look forward to seeing a revised version of your manuscript when it is ready. Please let me know if you have questions or comments regarding the revision.

Kind regards,

Deniz Senyilmaz Tiebe

Deniz Senyilmaz Tiebe, PhD
Senior Scientific Editor
EMBO Reports

Renaissance School of Medicine
Stony Brook University

Christopher W. Brownlee, Ph.D.
Assistant Professor of Pharmacological Sciences
Renaissance School of Medicine
Centers for Molecular Medicine 440
Stony Brook University, Stony Brook, NY 11794
Tel: 631-632-1593
christopher.brownlee@stonybrook.edu

Deniz Senyilmaz Tiebe, PhD
Senior Scientific Editor
EMBO Reports Journal
February 6, 2025

Dear Dr. Senyilmaz Tiebe,

I am writing to express my sincere gratitude for the time you spent reviewing and considering our manuscript for publication (EMBOR-2024-60684V2: Palmitoylated Importin α Regulates Mitotic Spindle Orientation Through Interaction with NuMA, P. Sutton, N. Mosqueda, C. Brownlee).

Thank you again for your help navigating the issue with reviewer one and excluding their review of the revised manuscript as previously discussed. We have addressed the reviewer's concerns and attached a point-by-point response below. Additionally, we have added a middle author, Natalie Mosqueda, as her help performing new experiments and revising the manuscript has been instrumental in meeting the resubmission deadline. Finally, we have submitted a possible cover image which Natalie captured of a developing *Xenopus* brain labeled for Nestin (a neuroprogenitor marker), microtubules, and DNA. If the image is considered for the cover we can provide more details.

Thank you again very much for your time and consideration,

Sincerely yours,

Christopher W. Brownlee, Ph.D.
Assistant Professor
Pharmacological Sciences
Renaissance School of Medicine
Stony Brook University

Point-by-point Reviewer Response

Referee #1:

Correct positioning of the mitotic spindle is crucial for asymmetric cell division during development, and in stem cells. In metazoans, this process is regulated by cortically anchored LGN/NuMA complexes, which tether dynein to the membrane to ensure accurate spindle placement. Sutton and Brownlee propose that membrane-anchored importin α , via palmitoylation, recruits NuMA to the membrane. They investigate this using inhibitors of importin α palmitoylation, examining its effects on spindle positioning and brain development in *Xenopus laevis*.

While the concept is new, the data presented are not entirely compelling. The evidence often does not consistently support the authors' claims. There are concerns with the microscopy images and analysis methods used. Moreover, the study's reliance on chemical inhibitors with minimal genetic validation weakens the overall conclusions. Below are some points for the authors to consider:

This reviewer has consistently provided nearly identical summaries of this manuscript to multiple journals. Despite our diligent efforts to address their concerns prior to resubmitting the paper to new journals, it appears that they have not conducted a thorough critical examination of the manuscript. Several of their concerns have been addressed within the revised manuscript, while others lack logical coherence and reference figures from an earlier manuscript

Major Points:

1. In Fig. 1, the authors claim importin α localizes to the membrane in DMSO-treated HCT116 cells. However, this is not evident from the images provided.

We assert that the membrane localization of importin α is evident both in the images and through the quantification provided. In previous reviews, this had referee had cited the lack of quantification was what made it unconvincing, now that we have provided quantification, they have chosen to disregard it.

Importin α appears localized to spindle poles and multiple punctate structures closer to the membrane. However, without a plasma membrane marker, this can't be concluded. The brightfield images in Fig. S2 are also unconvincing. I wonder why the authors did not use any plasma membrane marker to validate this observation. Also, in some images, in particular, cells treated with Wnt-C59, it appears that importin alpha could be a part of other membrane structures such as ER, and that is why it is probably present in the membrane fraction in their immunoblot analysis (Fig. 1E).

The localization of ER in mitotic cells differs significantly from the localization observed here for importin α . Furthermore, organelles, nuclei, and cytoplasmic membranes have been separated from the plasma membrane fraction as stated in line 176 of the manuscript.

Other than the plasma membrane marker, a colocalization of importin alpha with the ER marker will help better understand this localization pattern.

The localization of the ER during mitosis differs significantly from the localization observed here for importin α . Additionally, this study focuses on the proportion of importin α localized to the PM versus the cytosol. Consequently, we feel the use of an ER would not be of beneficial to the present study.

In general, the localization pattern at the membrane is not what has been observed for LGN/NuMA during metaphase (for instance, see Kiyomitsu and Cheeseman, 2012; Kotak et al., 2013; Seldin et al., 2013).

This is a comment from a review of an older version of the manuscript. We have addressed this misunderstanding in the current manuscript (lines 253-259, 304-308, 429-431) and in Figure 6. We do not expect the localization of importin α to match that of LGN/NuMA due to interactions with the RanGTP gradient precluding importin α from interacting with NuMA at the lateral cortex. This mechanism elegantly explains the long-reported and unique pattern of NuMA at the cortex during mitosis.

Also, some cells do not look like they are at the metaphase stage but instead in a prometaphase state (i.e., Palmostatin-treated cells).

Cells were arrested in metaphase with MG-132 treatment for 1 hour and so it is unlikely the cells would be found in pro-metaphase. Instead, it is possible that there could be chromatin enlargement due to increased importin α PM partitioning. Brownlee and Heald (2019) and Zhou et al. (2023) observed changes to nuclear size scaling when importin α was partitioned to the PM by palmostatin treatment. We may be observing a similar effect. This concern has now been addressed in the manuscript (lines 202-208):

2. In Fig. 2, the authors explore how these inhibitors impact spindle orientation. It remains unclear whether the effects are due solely to changes in importin α cortical localization. Given that Wnt signaling influences spindle positioning, the effects observed with WntC59 could be indirect. Although the authors used importin α -CAAX to rescue the WntC59 effect, importin α -CAAX cells already show spindle orientation defects.

We posit that a clear rescue is evident when importin α -CaaX is expressed in Wnt-C59 treated cells. Cells expressing importin α -CaaX exhibit resistance to Wnt-C59-induced spindle orientation defects, whereas cells expressing wild-type importin α remain susceptible to these defects. Moreover, cells expressing importin α -CaaX and treated with Wnt-C59 display misaligned spindles 50% less frequently than Wnt-C59 treatment alone. (see lines 230-239 and Figure II of the manuscript). These results strongly suggest that the lack of importin α palmitoylation following Wnt-C59 treatment is the primary cause of spindle misorientation.

It remains unclear whether it is because of excess/homogenous NuMA at the cell cortex.

The experiment in question utilizes HCT116 cells transiently expressing mClover-NuMA. As is evident in Figures 1 and 3, the DMSO control conditions demonstrate NuMA localization and spindle orientation matching that of HCT116 cells. Additionally, as the drug treatments exhibit significant spindle misorientation in comparison to the DMSO control condition, the specific use of HCT116 mClover-NuMA is irrelevant.

In Figure S2A, authors have costained NuMA with importing- α -CAAX; the images are of poor quality and non-conclusive: a mixture of mitotic and non-mitotic cells are imaged. I would suggest authors critically assess the relevance of cortical importin α in spindle orientation using some genetic approaches rather than primarily relying on inhibitors.

In lieu of genetic manipulation, we employ the validated inhibitor of PORCN, Wnt-C59, due to its ability to precisely control the timing of inhibition. This is particularly critical in the *in vivo* experiments, where morpholino or CRISPR knockdowns/knockouts of PORCN result in the complete loss of many tissue types due to the early requirement of Wnt signaling during development.

3. Next, the authors use a PLA to study importin α -NuMA interactions (Fig. 3).

The reviewer incorrectly asserts that the PLA assay is depicted in Figure 3, as the current manuscript submitted to EMBO includes this data in Figure 2. This discrepancy is another indication that the reviewer has not updated their previous reviews and has not critically examined the manuscript.

However, the claim that these interactions occur specifically at the polar cortex is not well-supported by the images.

The interactions occur within an ROI specifically labeled as “Polar Cortex” in Figure 2A. The methodology for quantifying these interactions is described on lines 241-245 of the manuscript, graphically represented in figure 2A, and outlined in the methods section of the manuscript.

The PLA lacks controls to verify whether the puncta are at or near the cell cortex. I wonder why the authors did not perform a simple colocalization experiment for NuMA and importin- α to see if they colocalize before performing the PLA experiment. They could further perform a confocal live imaging analysis for importin α and NuMA during mitosis to follow the spatiotemporal localization of these proteins in mitosis.

The PLA assay detects proteins that colocalize in a much more sensitive manner than the experiment described by the (only proteins that interact within <40nm are represented as a positive signal). Furthermore, the PLA assay enables the staining of numerous cells at different stages of mitosis and subsequent quantification of protein interactions under different experimental perturbations.

4. On page 14, the authors states that 'It has previously been shown that deletion of NuMA's NLS

causes mislocalization of NuMA away from the polar cortex (Okumura et al., 2018)'. This is a misleading statement. Okumura et al., 2018 in their Fig. 5, performed optogenetics experiments using deltaNLS mutant (to avoid dimerization with endogenous NuMA) in their optogenetically induced dynein localization experiment. Here, the rationale was not to assess NuMA localization in these overexpression conditions. As shown in Tsuchiya et al., 2021 (Fig. 1), NuMAdeltaNLS, or NuMAdeltaexon24, which lack NLS-NuMA, can be seen in the cell cortex. Therefore, I suggest authors thoroughly analyze and quantify cortical NuMA localization in these mutants.

In all of the panels in Okumura et al., 2018 within Figures 4 and 5, the NuMA deltaNLS construct as well as four other constructs lacking the NLS sequence of NuMA, are absent from the cell cortex, along with the ability of each construct to recruit dynein quantified. A subsequent paper by the same group (Tsuchiya et al., 2021 figure 1) presents a single image of NuMAdeltaNLS at the cell cortex without quantification. While the papers present conflicting data on NuMAdeltaNLS localization, we prioritize citing the paper that employs a broader range of constructs and quantifications over the paper that only presents an image of a single cell. Additionally, we obtained the NuMAdeltaNLS construct from the Tsuchiya et al., 2021 paper through Addgene and discovered that this construct was not the correct plasmid, but a different mutant of NuMA altogether, Addgene is now aware of this issue, and this finding now throws into question the correctness of the single cell image from the Tsuchiya et al, 2021 paper. Despite these concerns, we maintain our confidence in citing the Okumura et al. 2018 paper as presented in the current manuscript.

5. In Fig. 4, the authors examined the localization of NuMA at the cell cortex. As mentioned above, the cortical localization of importin α differs significantly from that of NuMA (punctate versus uniform cortical distribution).

It should be clarified that Figure 3 (not figure 4? Again this same mistake was made in a previous review) was imaged using a different microscope compared to Figure 1. The punctate appearance of importin α in Figure 1 can be attributed to deconvolution processes and not localization. Moreover, NuMA and importin α should not directly overlap (see lines 288-292, 338-340,498-506 of the manuscript).

This disparity raises questions about how the authors attribute NuMA's cortical localization to importin α .

This issue is addressed in the paper several times (lines 288-292, 338-340,498-506). It appears that this reviewer did not critically examine the manuscript, as we included these repeated clarifications on the localization of importin α vs NuMA specifically to address these concerns they raised in their previous review.

The authors claim that in cells treated with Palmostatin B, NuMA localization is no longer enriched at the polar cortex but distributed throughout the cortical membrane. However, their images do not clearly support this claim. It appears that NuMA is present at the lateral membrane at one side but not the other.

This feedback is from a previous review by this same reviewer, which was addressed by using a new representative image for the specific figure in question. Notably, this entire section (concern #5) and the next section (concern #6) is copied almost word for word from their previous review, despite our addressing these concerns. These duplicated sections are of particular concern, as this is a point that we specifically address within the manuscript, but has been disregarded by this reviewer. We have chosen new representative images for palmostatin B treatment to address this reviewer's concerns, yet their review remains unchanged.

6. The authors examine microcephaly and craniofacial defects after inhibitor treatment. While they suggest Wnt-C59 affects craniofacial development by inhibiting importin α palmitoylation, Wnt signaling itself plays a major role in stemness and growth, making it difficult to attribute these effects solely to importin α disruption. The high mortality rate and severe defects seen with Wnt-C59 strongly indicate a broader disruption of Wnt signaling.

This concern is addressed within the manuscript (see lines 446-452, 454-457, 520-524). Once more, this referee has demonstrated a lack of thoroughly examining the manuscript.

Partial rescue of PH3 with importin α -CAAX does not definitively clarify whether spindle positioning is restored or results from excessive importin α activity.

We have compared wild-type importin α against importin α -CaaX expression to address this concern in Figure II and J when analyzing spindle orientation and found no impact on spindle orientation when wild type importin α was overexpressed.

Expressing non-CAAX importin α could provide clearer insights, and assessing NuMA/dynein localization in importin α -CAAX cells would further aid understanding of these results.

As reviewer 3 has indicated, it is virtually impossible to discern the localization of NuMA/dynein within the *Xenopus* embryos, primarily due to the notoriously challenging nature of capturing and imaging the membrane interactions, even in established cell culture systems.

Minor Point:

-The authors' literature citations need improvement. For example, the statement on p. 3 (lines 50-51) regarding NuMA/dynein in chromosome segregation lacks vital references, such as Seldin et al., 2013; Zheng et al., 2013; Kotak et al., 2014. Similarly, Okumura et al., 2018 incorrectly cited LGN and G α 's role in NuMA localization during metaphase.

Kotak 2014, Seldin 2013 and Zheng 2013 have all been cited.

-The rationale for using HCT116 cells (p. 3) seems non-essential. HeLa, U2OS, and hTERT-RPE1 cells are widely used in spindle orientation studies, so stating the use of HCT116 for this study would suffice.

Referee #2:

General summary

The orientation of the spindle plays a crucial role by determining the position of the division plane. The authors report that the nuclear transport protein importin alpha when palmitoylated, plays a significant role in spindle orientation by regulating the localisation of another spindle rotation regulators (NuMA).

They observe craniofacial developmental defects in *Xenopus* lacking importin α palmitoylation: smaller head and brains. This is a frequent phenotype associated with spindle misorientation and is linked to microcephaly. These findings confirming a role for importin α in spindle orientation are useful, but they should cite past work on Importin alpha. The work also needs to significant improvement to figure presentations (suggestions provided below).

Major concerns that must be addressed

The manuscript is clearly written.

However the images presented in Figure 1 and Sup figure-2, do not clearly show spindle orientation defects. The team will benefit from recording some of the key experiments using live-cell microscopy. As the spindle oscillates and rotates, it transiently occupies incorrect positions and orientation.

While we agree that performing live cell analysis of cells as they progress through mitosis in the presence of our palmitoylation disrupting drugs could enhance the observation of spindle misorientation, we find this to be technically impractical. We believe that the fixed cell images and subsequent analysis are sufficient to demonstrate spindle misorientation, and we currently lack the necessary equipment to perform live cell spindle pole tracking. The method section details how spindle orientation defects are calculated from a z stack of the representative images (a method commonly used in the field). While it is true that the spindle oscillates from perpendicular to the substrate, quantifying the angle from perpendicular across hundreds of mitotic cells accurately recapitulates the oscillations seen from imaging one cell over the course of mitosis, with the added advantage of significantly increasing the number of cells analyzed.

Figure 2B and 2C are very unclear.

We have reviewed our images for this experiment and have determined that these are the most representative images for this data set. We have submitted individual figure files that are of a higher resolution to improve the clarity of these images.

DNA images in figure 3 do not appear normal. Importin alpha disruption can affect chromosome congression which can in turn cause spindle tumbling - so the authors have to be careful about explaining whether the spindle misorientation is a direct consequence of changes in cortical pathways or an indirect phenotype arising from congression defects (Fig 1B Wnt -C59 seems to affect chromosome alignment).

We appreciate the reviewers comments referencing the DNA images in Figure 3. Brownlee and Heald (2019) and Zhou et al. (2023) observed changes to nuclear size scaling and chro-

matin scaling when importin α was partitioned to the PM through palmostatin treatment. We speculate that a similar effect may be observed in this study and have added a discussion of this phenomenon to the manuscript (lines 202-208). Additionally, the finding that importin α - CaaX but not Wt expression rescued the spindle orientation defects after the same perturbations (Figure 1I), we feel confident in our explanation that the spindle misorientation defects are arising from a lack of importin α at the cortex.

Another way to showcase cortical role would be to ask whether changes to NuMA affect downstream events (eg., localisation of DHC)? Changes in cortical localisation of NuMA are slightly unclear. DHC-GFP or dynein localisation studies may help confirm the impact of changes in NuMA.

We appreciate the reviewer's suggestion and have conducted a new experiment investigating the effects of importin α palmitoylation disruption on dynactin localization. This analysis aims to elucidate the downstream consequences of NuMA mislocalization (New Figure 3E-H). The description of this experiment has been incorporated into the manuscript (lines 359-382).

Figure 4 phenotypic differences are clear and well presented.

In *Xenopus* and human cells, the long-axis of interphase cell is known to predetermine the position of the spindle.

While it is true that physical and biochemical inputs regulate spindle orientation upstream of NuMa localization, this study is aimed at dissecting the mechanism by which the spindle orientation machinery is connected to the cell cortex in a RanGTP and importin-dependent manner. However, we have added a discussion of different physical properties affecting spindle orientation to the manuscript (lines 248-252):

LGN plays a key role in defining the division-plane in relationship to the long-axis (Zulkipli et al., JCB 2019). Would Importin alpha play a role in linking cell shape and spindle orientation? Are the authors able to discuss or assess interphase cell shape in their data?

We appreciate the reviewer's suggestion and conducted an analysis of the impact of importin α palmitoylation disruption on cell shape (new Figure EV2B) The findings of this analysis are discussed in the manuscript (lines 252-258).

Minor points

Typo in sup figure 3 title.

We thank the author for the suggestion but we were unable to identify a typo in this figure title. Sup figure 4 legend requires explanation.

The figure legend for this figure has been expanded to provide more detail.

Figure 2E requires statistical significance

The data presented in Figure 2E, now Figure 2F, is all nonsignificant. In order to make this more apparent we have added a description of the figure's statistical significance to the figure legend:

Missing key citations:

Importin α phosphorylation promotes TPX2 activation by GM130 to control astral microtubules and spindle orientation

Guo et al. J Cell Sci. 2021.

This is a highly relevant paper that we have added to the paper and cite when discussing previous research on importin α functioning outside of nuclear transport.

Mechanics of spindle orientation in human mitotic cells is determined by pulling forces on astral microtubules and clustering of cortical dynein

Anjur Dietrich et al., Dev Cell. 2024

This paper is relevant (although not published when this manuscript was submitted) and has now been cited when we discuss aMT anchoring.

Referee #3:

Sutton and Brownlee hand in their work on palmitoylated Importin α regulating mitotic spindle orientation through interaction with NuMA. The authors continue to work on the idea that palmitoylation of Importin α regulates its localized interaction with client, i.e. NLS-containing, proteins in mitosis at the plasma membrane and modulates key functions, i.e. here: spindle orientation. They further focus on NuMA whose liberation after NEB in open mitosis puts it back under the control of importin.

The manuscript combines experiments in human cells with approaches in developing *Xenopus* embryos to stretch the bow from molecular hypotheses to cellular and finally to organismal observations. To my opinion, it is that range of experiments that could justify publication and I am therefore, in principle, in favor of publishing the work in EMBO J. However, there are several issues that I find would need to be addressed prior to publication. I focus most details (major and minor summarized along the individual experiments) on the cellular observations where my expertise is more useful than in the *Xenopus* embryo experiments the technical quality of which find hard to judge in detail.

In the first set of experiments, Sutton and Brownlee measure spindle orientation in human cells (HCT116) in culture and use small molecule effectors to either inhibit or to stimulate palmitoylation of importin. Alongside, the interaction with or release of NLS-proteins to and from importin is inhibited.

Fig. 1 uses small molecule inhibitors and reveals a significant defect of importin localization correlated to spindle misorientation when inhibiting palmitoylation. Fair enough, but (1) only one inhibitor is used here, (2) statistics are apparently from a single experiment and (3) the negative results come without controlling that the other inhibitors do work effectively under these conditions. While argument (3) may only require a somewhat toned-down wording, (1) and especially (2) need to be addressed experimentally.

We thank the reviewer for the comments but are somewhat confused as (1) says only one inhibitor is used, yet four different drug inhibitors are used. As mentioned in (2) the data is from a single experiment but is carried out with multiple replicates. We have rewritten the figure legend to clarify these points.

The immunoblot in Fig. 1E is important, why not comparing with conditions of palmitoylation inhibition/stimulation?

We thank the reviewer for suggesting this experiment. We have attempted to repeat this experiment with expanded conditions, but unfortunately encountered a number of technical issues regarding a loss of efficacy in our palmitoylation disrupting drugs. After repeated attempts at this experiment we found a decrease in importin α signal at the PM when palmitoylation was disrupted although upon repeating and quantifying this result was variable and not statistically significant. Due to the decreased efficacy in our drug array most likely due to age or expiration, additional drug stocks needed to be reordered and will require additional time to retry the experiment. In order to submit this revised manuscript in a timely fashion and before the re-

submission deadline we have chosen to not include the repeated experiments with the diminished efficacy drug stocks.

Fig. 1I addresses a key argument, i.e. rescue of palmitoylation inhibition using independently plasma membrane targeted importin. I think this should be included with a representative image into Fig. 1G.

We thank the reviewer for their suggestion and have added a new panel to figure 1 (new Figure 1J) with representative images for HCT116 cells transfected via nucleofection with importin α -mCherry-HA-CaaX and treated with either DMSO or Wnt-C59. These results demonstrate that there is no observable difference in spindle orientation between DMSO and Wnt-C59 treated cells when expressing importin α -mCherry-HA-CaaX.

Fig. 2. Contains an intriguing approach to address the localization of importin-regulated NLS proteins, i.e. NuMA and DLG. Using Duo-link PLA, the specific interactions are visualized. However, how is this controlled? Could the reduced interaction at the PM be rescued using the CAAX-construct like in Fig. 1?

We thank the reviewer for suggesting this valuable experiment and we have performed a rescue experiment for the PLA similar to the one performed in Figure 1, using a CaaX construct (new Figure 2G and H) which is discussed in the manuscript (lines 303-313).

The image in Fig. 2 B, center, shows reduced overall PLA signal, why is that? Or is this just not really representative?

We thank the reviewer for bringing this interesting observation to our attention. After analyzing the images, we posit these differences might be attributed to the effects of the mitotic Ran gradient interacting with importin α/β . In the Wnt-C59 treatment we have demonstrated that importin α is displaced from the PM and is enriched in the cytoplasm (Figure 1A-D). Additionally, in mitosis there is an abundance of RanGTP at the center of the cell due to the mitotic Ran gradient which prevents importin α from binding to NLS containing cargo such as NuMA. We hypothesize that in Wnt-C59 treated mitotic cells there would be a large shift of importin α into the cytoplasm where it would no longer be able to bind NuMA resulting in this drop of intensity of PLA signal that we see in the mitotic Wnt-C59 treated cells, but not the interphase Wnt-C59 treated cells, where the RanGTP gradient is contained within the nucleus.

The IP-Western experiment in Fig. 2F does not contain a negative control either.

We thank the reviewer for identifying this shortcoming and the IP-Western has been repeated with GAPDH as a negative control (new Figure 2A).

Fig. 3 addresses the localization of NuMA under different conditions of importin localization to the PM. The authors use a fluorescently tagged transiently overexpressed NuMA variant. Variable expression levels are normalized cell-internally in the quantification, however, expression levels may affect the distribution on their own. Has this been addressed?

We thank the reviewer for this comment. Although we have not directly addressed the NuMA expression levels in this experiment, we do not predict it would have any impact on NuMA localization. Additionally, overexpression is necessary in order to effectively visualize NuMA localization at the cell cortex. We have included an acknowledgement of this in the manuscript and emphasized that the DMSO control condition exhibits NuMA localization that aligns with previously reported work (lines 333-336).

I don't see any indication of the sample size to which statistics are based on.

For figure 3 the sample size is stated in the figure legend ($n > 40$).

How does this assay compare to a plain indirect immunofluorescence detecting endogenous NuMA? The latter would also address if endogenous NuMA follows the exogenously expressed Clover-fusion.

We thank the reviewer for this comment. The mClover-NuMA localization we observe in our control cells matches what has been previously seen with endogenous NuMA. We utilize the mClover-NuMA expression to better visualize NuMA at the polar cortex as there is only a very small amount of NuMA actually present and without this transient expression the signal present at the cortex is often washed out by the signal at the centrosomes. By using this transiently expressing mClover-NuMA we can more accurately visualize NuMA changes at the cell cortex.

How do the authors explain that palmostatin strongly influences NuMA mitotic PM distribution but does not affect spindle orientation as shown in Fig. 1? Did I miss anything here?

We hypothesize that a complete lack of NuMA at the cell cortex as seen in Wnt-C59 conditions has a more severe effect on spindle orientation than NuMA localizing to both the polar and lateral cortices as seen in palmostatin conditions. We have added a description of this to the manuscript (lines 348-358).

Figs. 4 switches gears and starts to address a potential function of importin regulating NuMA in developing *Xenopus* embryos. Again, palmostatin does not do much; given what we have seen in Fig. 3, why is that? It is certainly technically very difficult/impossible to localize endogenous NuMA in frog embryos, but the correlation should at least fit with what is seen in cell culture (Fig. 3).

We feel that Figure 4 does in fact show a significant effect of palmostatin on *Xenopus* head/craniofacial development (lines 407-409) and on brain development (lines 423-428).

We also feel that it would be impractical to analyze endogenous mitotic NuMA localization in a developing *X. laevis* embryo. To conduct this experiment, we would require use of PFA fixed embryos immunostained for NuMA and subsequently scan the embryo brain for a cell, that by chance, is in metaphase at the exact moment of fixation, in an orientation that would allow us to analyze NuMA localization in a method akin to Figure 3. Additionally, the identification of such a cell and the detection of sufficient cortical NuMA to facilitate these quantifications, particularly given only endogenous NuMA levels in an environment with a high degree of back-

ground signal due to the quantity of cells in the developing brain, renders this suggested experiment exceptionally challenging.

Fig. 5: The experiments' highlight seems to be the partial rescue of an otherwise lethal phenotype when applying WntC59 to frog embryos. However, the analysis remains superficial as it just uses cell division (phospho H3) as readout. The quantification in Fig. 5D comes without any further specification of the sample size used here.

While figure 5 relies on cell division as a readout, we believe that this is a valid and meaningful metric for quantifying brain development in the context of microcephaly. Microcephaly can be characterized by premature differentiation of neuroprogenitor cells which would in turn result in a depletion of the number of actively dividing cells in the developing brain. Additionally, there is a sample size listed for Figure 5D in the figure legend ($n > 10$).

Further issues:

I feel that the flow of logical arguments in the introduction is somewhat difficult to follow: putting importin α into the limelight of an abundant (this arguments may be included!) adaptor whose localization at the plasma membrane targets NLS proteins throughout mitosis is very reasonable and consistent with fading out of the Ran gradient in the cell periphery. So, the Ran arguments should go first or in parallel with the evidence for palmitoylation as a targeting mechanism.

We thank the reviewer for his feedback on the introduction section. We have added a comment on the abundance of importin α in cells to the introduction, as suggested.

The discussion claims that importin has been solely attributed to nuclear import. It is certainly correct to say that the membrane targeted form reveals a new function in mitosis. However, importin α has been initially characterized as the NLS receptor, being instrumental for nuclear import but also in a chaperone like function in the interphase cytoplasm and as a regulator of NLS protein accessibility in mitosis. The "new" function should be clearly sorted under this general concept (as, in fact, it is in the introduction).

We thank the reviewer for their comment and have updated the description of importin α function in the discussion to match the one in the introduction (lines 486-49).

Suppl. Figs 4 and 5 miss indication of scales.

Scale bars have now been added to these figures to address this.

Dear Chris,

Thank you for submitting your revised manuscript. It has now been seen by one of the original referees.

As you can see, referee finds that the study is significantly improved during revision and recommend publication. However, I need you to address the points below before I can accept the manuscript.

- Please provide 3-5 keywords for your study. These will be visible in the html version of the paper and on PubMed and will help increase the discoverability of your work.
- Please rename the Data and Code Availability section as Data Availability section.
- Please deposit the microscopy data mentioned in the Data and Code Availability section into a public repository such as BioImage Archive and add a link directly resolving to the dataset in this section. Please see for <https://www.embopress.org/page/journal/14693178/authorguide#datadeposition> further details.
- Please rename the 'Declaration of Interests' section as 'Disclosure And Competing Interests Statement'.
- Please remove the 'Author Contributions' section from the manuscript text.
- As per our format requirements, in the reference list, citations should be listed in alphabetical order and then chronologically, with the authors' surnames and initials inverted; where there are more than 10 authors on a paper, 10 will be listed, followed by 'et al.'. Please see <https://www.embopress.org/page/journal/14693178/authorguide#referencesformat>
- We note that there are three citations to preprints in the reference list: Camuglia et al, 2022, Fankhaenel et al, 2023 and Neville et al, 2022, which need to be cited according to the preprint citation format. Below is an example:
Citations to manuscripts posted on recognized preprint servers can be cited the following way:

In-text citation: (preprint: NAME1 et al, YEAR)

Reference list: Author NAME1, Author NAME2, (YEAR) article title. bioRxiv doi: 1234/002.dfg123 [PREPRINT]

- We note that B4, B5 and B6 cells were not filled in in the author checklist.
- Please remove the Reagents and Tools table from the manuscript text and submit it as a separate word file.
- In the source data for Figure 2A, the annotation boxes for the NuMA and GAPDH blot seem to be misplaced. Please double check.
- Please rename the Materials and Methods section as Methods.
- The manuscript sections should be in the following order: Title page - Abstract & Keywords - Introduction - Results - Discussion - Methods - Data Availability - Acknowledgments - Disclosure Statement & Competing Interests - References - Figure Legends - (Main Tables with legends if applicable) - Expanded View Figure Legends.
- We note that nomenclature for EV figure legends needs to be updated as follows: it should be Figure EV1, etc. instead of Expanded View Figure 1, etc.
- During our routine check, we note possible image reuse at two instances in Figure EV4. Please clarify.
- Our production/data editors have asked you to clarify several points in the figure legends - Figure Legends (main + EV):
 - o Please note that the exact p values are not provided in the legends of figures 1B, C, D, H, J, L; 2D, H; 3B, C, D, F, G; 4B, D; 5D, EV2 B
 - o Please note that information related to n is missing in the legends of figures 2J, K
 - o Please note that the error bars are not defined in the legends of figures 2J, K.
- Papers published in EMBO Reports include a 'synopsis' and 'bullet points' to further enhance discoverability. Both are displayed on the html version of the paper and are freely accessible to all readers. The synopsis includes a short standfirst summarizing the study in 1 or 2 sentences (max 35 words) that summarize the paper and are provided by the authors and streamlined by the handling editor. I would therefore ask you to include your synopsis blurb and 3-5 bullet points listing the key experimental findings.
- In addition, please provide an image for the synopsis. This image should provide a rapid overview of the question addressed in the study but still needs to be kept fairly modest since the image size cannot exceed 550 (width) x 300-600 (height) pixels.

Thank you again for giving us to consider your manuscript for EMBO Reports, I look forward to your minor revision.

Kind regards,

Deniz

--

Deniz Senyilmaz Tiebe, PhD
Senior Scientific Editor
EMBO Reports

Referee #3:

Sutton, Mosqueda and Brownlee hand in a revised version of the manuscript entitled "Palmitoylated Importin alpha Regulates Mitotic Spindle Orientation Through Interaction with NuMA" that was previously reviewed for EMBO J but now is targeted, in its revised form, to EMBO R. Even though I the set-up of the research question and the complexity of observations leaves open a number of questions, I stay with my previous general evaluation that the work should be published (now) in EMBO R due to its large scope and courageous approach to analyze a potential function of PM-localized importin alpha to regulate downstream proteins (NuMA) in mitosis. Having said this, I see that the authors argue (in a positive sense) with most of my comments, while experimentally addressing only some of them, fair enough. Commenting on and explaining apparent differences in phenotypic observations between Wnt-C59 and palmostatin clarifies this issue. I also appreciate the rescue experiment with CAAX- importin in Fig. 2G and H. Taken together, the pendulum swings in favor for publication and I suggest accepting the manuscript in its present form.

All editorial and formatting issues were resolved by the authors.

Dr. Christopher Brownlee
Stony Brook University
Pharmacological Sciences
100 Nicolls Road
Centers for Molecular Medicine 440
Stony Brook, New York 11733
United States

Dear Chris,

Thank you for submitting your revised manuscript. I have now looked at everything and all is fine. Therefore, I am very pleased to accept your manuscript for publication in EMBO Reports.

Congratulations on a nice work!

Kind regards,

Deniz

--

Deniz Senyilmaz Tiebe, PhD
Senior Scientific Editor
EMBO Reports
